# Mitigating Gradient Pathology in PINNs through Aligned Constraint

**Yichen Luo** [1]  **Peiyu Zhu** [2]  **Dongxiao Hu** [3]  **Jia Wang** [3]  **Tailin Wu** [4]  **Dapeng Lan** [5]  **Yu Liu** [5]  **Zhibo Pang** [2]

## Abstract

While Physics-Informed Neural Networks (PINNs) are powerful for solving Partial Differential Equations (PDEs), their training is often paralyzed by gradient pathology. The gradients from the PDE residuals and boundary constraints oppose each other, trapping the model in local minima. Current solutions, such as adaptive weighting or hard constraints, either fail to fundamentally resolve this ill-conditioning or are limited to simple geometries. In this study, we systematically analyze the possible causes of this gradient pathology from the perspectives of loss landscapes and optimization dynamics. Based on the obtained conclusion, we propose Constraint-Aligned loss with Manifold Lifting (CAML). By reformulating all zeroth-order terms into aligned constraints, our method effectively mitigates gradient conflicts. In addition, we introduce a delay factor to help the optimizer skip the high-curvature area. Experiments demonstrate that our CAML significantly enhances numerical stability and efficiency in highly complex PINN problems. Our code is open-sourced on CAML.

## 1. Introduction

Physics-Informed Neural Networks (PINNs) (Raissi et al., 2019) have established themselves as a transformative framework in scientific computing, offering a robust deep learning strategy for solving Partial Differential Equations (PDEs). By embedding PDE residuals and boundary conditions directly into the loss function, this approach signif-

icantly mitigates the dependence on labeled training data. Notably, PINNs that rely exclusively on physical prior functions can be considered an innovative meshless numerical method. This paradigm enables the completely self-supervised reconstruction of target physical fields, yielding neural models that generalize effectively without the need for costly data acquisition. In domains such as linear elasticity (Haghighat et al., 2021; Samaniego et al., 2020), fluid mechanics (Raissi et al., 2020; Jin et al., 2021), heat conduction (Cai et al., 2021; Haddout & Haddout, 2025), and electromagnetism (Chen et al., 2020; Lim & Psaltis, 2021), such a well-trained model is poised to supersede traditional numerical methods, providing nearly real-time solutions for previously computationally-intensive geometric problems that require an extremely long period of time.

However, compared to traditional data-driven paradigms, employing PINNs to approximate PDE solutions remains non-trivial. A salient limitation is the ill-conditioning of the loss function, which severely hinders convergence, even in relatively elementary scenarios (Krishnapriyan et al., 2021). Literature indicates that the gradients derived from the PDE residuals and the boundary constraints often exhibit conflicting directions, causing the optimization process to be trapped in local minima (Wang et al., 2021; 2022b; Rathore et al., 2024). Consequently, in problems involving complex geometries or PDE configurations, this pathology prevents the model from accurately recovering the ground truth.

Recent work has addressed this challenge from several distinct perspectives. For example, Yu et al. (2022); Wang et al. (2022b) and Park et al. (2024) focus on designing adaptive loss functions to equilibrate the pathology between boundary conditions and PDE residuals. In contrast, Müller & Zeinhofer (2023); Hwang & Lim (2024) and Liu et al. (2025) attempt to enhance the optimizer by dynamically adjusting the optimization trajectory to navigate the non-convex landscape. Another line of work, Wang et al. (2024) and Dasht-bayaz et al. (2024), enhances performance through a unique configuration of the model. Nevertheless, these methods mainly alleviate convergence issues stemming from gradient ill-conditioning, without fundamentally mitigating the ill-conditioning itself. Alternatively, model-based improvements, such as Sun et al. (2020) and Sukumar & Srivastava (2022), eliminate the boundary loss term by enforcing hard constraints. However, such methods typically rely on strict

---

[1]Department of Information Science and Engineering, KTH Royal Institute of Technology, Stockholm, Sweden [2]School of Advanced Manufacturing and Robotics, Peking University, Beijing, China [3]School of Advanced Technology, Xi'an Jiaotong-Liverpool University, Suzhou, China [4]Department of AI, School of Engineering, Westlake University, Hangzhou, China [5]Techforgood AS, Oslo, Norway. Correspondence to: Zhibo Pang <zhibo.pang@pku.edu.cn>.

*Proceedings of the 43$^{rd}$ International Conference on Machine Learning*, Seoul, South Korea. PMLR 306, 2026. Copyright 2026 by the author(s).

Dirichlet boundary conditions and distance functions, which severely restrict their generalizability to complex geometries and combined boundary conditions.

In this work, we provide the explicit geometric characterization of the PDE-residual loss landscape in PINNs. We show that operator non-uniqueness induces a connected manifold of global minimizers for the PDE residual, forming an ill-conditioned valley in parameter space and intrinsically causing gradient conflicts with boundary losses. Based on this insight, we propose **C**onstraint-**A**ligned loss with **M**anifold **L**ifting (**CAML**), which directly modifies the geometry of the optimization landscape to mitigate gradient pathology. Our contributions are summarized as follows:

1. We formally prove that the PDE residual loss admits a connected set of global minimizers in function space, and show how this manifold induces possible gradient conflict after mapping to parameter space.

2. We reformulate all zeroth-order terms as explicitly solvable aligned constraints, enlarge the intersection between PDE and boundary admissible sets, thereby fundamentally reducing gradient inconsistency.

3. We introduce a delay factor for residual loss to prevent premature descent into unfavorable regions of the residual-induced valley, thus shortening the optimization trajectory and improving stability.

Experiments across various physical fields demonstrate that these innovative methods significantly enhance the stability and efficiency of the model when solving geometrically complex problems or nonlinear boundary conditions.

**Conflict of Interest.** The authors declare that there are no financial or other conflicts of interest related to this work.

## 2. Related Work

### 2.1. General Optimization Strategies

A series of studies has systematically investigated the pathology of PINN loss functions from multiple perspectives, including gradient conflicts between PDE residuals and boundary conditions (Wang et al., 2021), the ruggedness of the loss landscape (Krishnapriyan et al., 2021; Rathore et al., 2024), and derivative pathologies (Park et al., 2024; Song et al., 2024). These challenges have spurred the development of various mitigation strategies, including model architectures, loss formulations, and optimization methods.

Loss function modifications provide the most direct remedy. Representative approaches include approaches that incorporate gradient information of PDE residuals to regularize steep solution landscapes (Yu et al., 2022), adver-

sarial $L^\infty$-based PINNs for high-dimensional Hamilton-Jacobi-Bellman (HJB) equations (Wang et al., 2022a), meta-learning formulations (Psaros et al., 2022), neural spectral methods (Du et al., 2024), and auxiliary-variable decompositions for high-order derivatives (Park et al., 2024). In addition, there are a series of methods (Zhang et al., 2024; Chen et al., 2025; Gao et al., 2025; Zhou et al., 2025a) that provide implicit regularization through adaptive weights.

Optimizer-based methods attempt to navigate in the non-convex landscape more effectively. DCGD (Hwang & Lim, 2024) and ConFIG (Liu et al., 2025) dynamically adjust the optimization trajectories to maintain consistency between update directions and individual loss gradients. Müller & Zeinhofer (2023) leverage Hessian-induced Riemannian metrics to exploit curvature information. Rathore et al. (2024) and Boudec et al. (2025) propose optimizers that target pathology induced by differential operators.

Architecture modifications provide implicit regularization. PirateNets (Wang et al., 2024) introduce adaptive residual connections to stabilize PDE residual minimization under suboptimal initialization. Dashtbayaz et al. (2024) analyzes critical points through a wide neural network theory. Anand-kumar et al. (2020); Li et al. (2021); Dos Santos et al. (2023) and Zhao et al. (2024) attempted to enhance the prediction accuracy by using advanced architectures.

Despite these advances, existing methods primarily balance gradient magnitudes or adjust optimization trajectories without fundamentally eliminating the directional conflicts between boundary and interior loss terms. The underlying ill-conditioned loss landscape persists.

### 2.2. Domain-Specific Hard Constraint Methods

Another class of methods (Sun et al., 2020; Liu et al., 2022) enforces the calculation of boundary conditions through an ansatz-based hard constraint. By removing boundary terms from the objective, these methods fundamentally avoid gradient conflicts. Sheng & Yang (2021); Lu et al. (2021); Sukumar & Srivastava (2022); Xie et al. (2023) and Zhou et al. (2025b) further expanded on this and applied this hard constraint method to various fields. Nevertheless, these methods demand careful construction of distance functions and geometry-aware representations for each specific domain, posing significant challenges for generalization to arbitrary complex geometries.

### 2.3. Training Enhancements

Beyond gradient-based techniques, several optimization-oriented strategies have been proposed to improve convergence efficiency. Curriculum learning (Krishnapriyan et al., 2021) reduces training difficulty by progressively organizing samples, while adaptive sampling (Lau et al., 2024) dynam-

ically reallocates collocation points to enhance efficiency. Complementarily, recent work has analyzed PINN training failures from a gradient-dynamics perspective: Wang et al. (2021) employed the annealing algorithm to balance the interactions among different terms in the loss function, and Wang et al. (2022a) further formalized this phenomenon via neural tangent kernel analysis, proposing adaptive weighting to balance convergence across loss components. While orthogonal to gradient-aware approaches and easily integrated as auxiliary components, these enhancements do not directly resolve the fundamental conflict between boundary conditions and PDE residual gradients.

# 3. Preliminaries and Problem Setup

We consider a general steady-state PDE of the form

$$\mathcal{N}[u](x) = f(x), \quad x \in \Omega, \tag{1}$$

where $u$ denotes the unknown solution, $\mathcal{N}$ is the differential operator defining the PDE, $f$ is the source term, and $\Omega$ is the computational domain. To fully specify a problem instance, boundary conditions are imposed on $\Gamma = \partial\Omega$, which may consist of Dirichlet, Neumann, and Robin components:

$$
\begin{aligned}
u &= g_{\mathrm{d}}(x), & x \in \Gamma_{\mathrm{d}}, \\
\nabla u \cdot \mathbf{n} &= g_{\mathrm{n}}(x), & x \in \Gamma_{\mathrm{n}}, \\
\alpha u + \beta \nabla u \cdot \mathbf{n} &= g_{\mathrm{r}}(x), & x \in \Gamma_{\mathrm{r}},
\end{aligned}
\tag{2}
$$

where $\mathbf{n}$ denotes the normal outward unit vector, and $g_{\mathrm{d}}$, $g_{\mathrm{n}}$, and $g_{\mathrm{r}}$ are prescribed boundary value functions. In the Robin condition, $\alpha$ and $\beta$ are coefficients of the zeroth- and first-order terms, respectively.

For problems with composite boundary conditions, the standard PINN loss is defined as

$$\mathcal{L}(\theta) = w_{\mathrm{res}}\mathcal{L}_{\mathrm{res}} + w_{\mathrm{bc}}\mathcal{L}_{\mathrm{bc}}, \tag{3}$$

where $w_{\mathrm{res}}$ and $w_{\mathrm{bc}}$ balance the PDE residual and boundary condition losses. The PDE residual loss is given by

$$\mathcal{L}_{\mathrm{res}} = \frac{1}{N_{\mathrm{res}}}\sum_{i=1}^{N_{\mathrm{res}}} |\mathcal{N}[u_\theta](x_i) - f(x_i)|^2, \tag{4}$$

where $\{x_i\}_{i=1}^{N_{\mathrm{res}}} \subset \Omega$ are interior collocation points and $u_\theta$ denotes the model approximation parameterized by $\theta$.

The boundary condition loss $\mathcal{L}_{\mathrm{bc}}$ is defined as the sum of individual boundary losses, $\mathcal{L}_{\mathrm{bc}} = \mathcal{L}_{\mathrm{d}} + \mathcal{L}_{\mathrm{n}} + \mathcal{L}_{\mathrm{r}}$, with

$$
\begin{aligned}
\mathcal{L}_{\mathrm{d}} &= \frac{1}{N_{\mathrm{d}}}\sum_{i=1}^{N_{\mathrm{d}}} |u_\theta(x_i) - g_{\mathrm{d}}(x_i)|^2, \\
\mathcal{L}_{\mathrm{n}} &= \frac{1}{N_{\mathrm{n}}}\sum_{i=1}^{N_{\mathrm{n}}} |\nabla u_\theta(x_i) \cdot \mathbf{n}_i - g_{\mathrm{n}}(x_i)|^2, \\
\mathcal{L}_{\mathrm{r}} &= \frac{1}{N_{\mathrm{r}}}\sum_{i=1}^{N_{\mathrm{r}}} |\alpha u_\theta(x_i) + \beta \nabla u_\theta(x_i) \cdot \mathbf{n}_i - g_{\mathrm{r}}(x_i)|^2,
\end{aligned}
\tag{5}
$$

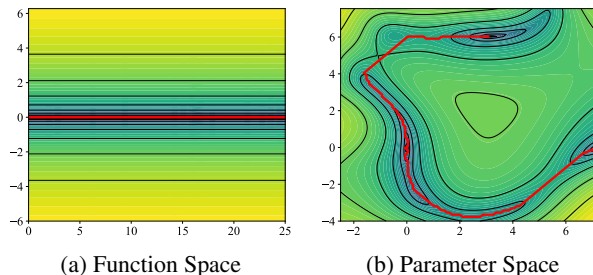

(a) Function Space      (b) Parameter Space

*Figure 1.* Visualization of the PDE residual loss landscape in both the function space and the parameter space, projected onto a 2D subspace. The red curve indicates the loss valley. While the landscape is relatively simple in the function space, it becomes highly distorted and non-convex when mapped to the parameter space by the neural network. More details are in Appendix G.

where $\{x_i\}$ are collocation points sampled on $\Gamma_{\mathrm{d}}$, $\Gamma_{\mathrm{n}}$, and $\Gamma_{\mathrm{r}}$, respectively.

Recent studies (Wang et al., 2021) have shown that the gradient directions associated with the PDE residual, $\nabla_\theta \mathcal{L}_{\mathrm{res}}$, and those induced by boundary conditions, $\nabla_\theta \mathcal{L}_{\mathrm{bc}}$, often exhibit a negative inner product. This gradient conflict leads to competing optimization trajectories and can significantly impede convergence.

# 4. Theoretical Analysis

## 4.1. Loss Landscape Perspective

Previous studies (Li et al., 2018; Rathore et al., 2024) have primarily investigated the geometry of the loss landscape in the parameter space. In this section, we instead adopt a complementary perspective by simultaneously considering the loss geometry in both the **function space** and the **parameter space**. Let $\mathcal{F}(\Omega)$ denote an appropriate Sobolev space defined in the domain $\Omega$. We define the loss functional in the function space as

$$\mathcal{L}_{\mathrm{func}} : \mathcal{F}(\Omega) \to \mathbb{R}, \ u \mapsto \mathcal{L}_{\mathrm{func}}(u) = \|\mathcal{N}[u] - f\|_{\mathcal{L}^2(\Omega)}^2. \tag{6}$$

Unlike the complete loss function defined in the parameter space that incorporates boundary constraints in Equation (3), this equation is a theoretical functional for the PDE residual.

### 4.1.1. PDE RESIDUAL TERM

**Theorem 4.1** (Existence of Loss Valleys in PDE Residual Term). *If the PDE constraint $\mathcal{N}[u] = f$ does not uniquely determine a solution in function space, then the PDE residual loss admits a flat and connected set $\mathcal{S}_{\mathrm{res}} = \{u \in \mathcal{F}(\Omega) \mid \mathcal{N}[u] = f\}$ of global minimizers. When represented by an over-parameterized neural network, this set induces a connected manifold of zero residual solutions in parameter space with at least one flat direction.*

*Table 1.* Hessian-based analysis of the loss landscape under three different additive constants in both function and parameter space.

| Const. | Final Loss | Condition Number | | Subspace Similarity | |
|---|---|---|---|---|---|
| | | **Func. Space** | **Param Space** | **Func. Space** | **Param Space** |
| 0 | 3.97e-7 | $\infty$ | $1.50 \times 10^{23}$ | – | – |
| 1 | 1.07e-5 | $\infty$ | $5.97 \times 10^{22}$ | 100.0% | 50.8% |
| -1 | 5.24e-8 | $\infty$ | $1.05 \times 10^{24}$ | 100.0% | 44.8% |

We provide the strict proof in Appendix A. This characteristic reflects the intrinsic non-uniqueness of the differential operator $\mathcal{N}$, which is parameterized by a small number of free modes in classical PDEs. Although the zero-residual set may correspond to a simple hyperplane in the function space, its preimage in the parameter space is generally mapped to a highly distorted and non-convex manifold due to the strong nonlinearity of this mapping, as visualized in Figure 1.

This characteristic is fundamentally distinct from the Mean Squared Error (MSE) constraint commonly encountered in classic regression problems, where its objective function admits only a single global optimum. In contrast, the continuous zero-residual set $\mathcal{S}_{\text{res}}$ of PINN loss yields a 'valley-like' loss landscape in the function space. The bottom of this loss valley consists of multiple distinct functions that exactly satisfy the governing equation, yet differ substantially in their absolute values. Consequently, unlike the 'flat minima' widely discussed in supervised regression contexts (Keskar et al., 2017; Garipov et al., 2018; Foret et al., 2021), this loss valley originates from the intrinsic non-uniqueness of the PDE solution under insufficient constraints, rather than the redundancy of network parameters. Importantly, even when the PDE loss is nearly zero, a significant optimization trajectory may still be required to reach the physically meaningful solution selected by the boundary conditions.

Rathore et al. (2024) provided a quantitative analysis of the PINN loss landscape by estimating the Hessian spectral density, revealing severe pathology with large outlier eigenvalues and a high density of near-zero eigenvalues, consistent with our loss valley hypothesis. Following their work, we further analyze the Hessian in both function and parameter space at converged solutions, as shown in Table 1. We observe that the zero-loss solutions in the function space exhibit an infinite Hessian condition number, confirming the existence of the strict zero-residual manifold predicted by Theorem 4.1. Furthermore, the subspace similarity in the function space reaches 100%, verifying the linear solution set shown in Figure 1. After being mapped onto the parameter space, this manifold forms a sharp valley, with a condition number over $10^{22}$. In addition, the low-curvature eigenspaces in parameter space only partially align across different additive constants ($\approx 50\%$), indicating that the loss valley forms a broad and geometrically complex manifold, rather than a fixed-dimensional mode. More details about this experiment are in Appendix G.

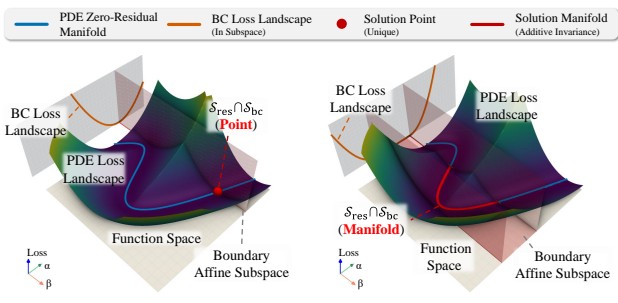

(a) Unique (Well-Posed)  (b) Non-Unique (Ill-Posed)

*Figure 2.* Schematic illustration of the intersection between PDE residual loss landscape and boundary-condition constraint landscape in **function space**. The zero-residual solutions of the governing PDE form a manifold, while boundary conditions restricting the admissible solutions to a subset. Their intersections correspond to valid solutions, which can be (a) unique or (b) non-unique.

### 4.1.2. BOUNDARY CONDITION TERMS

Boundary conditions can be viewed as constraints in the function space $\mathcal{S}_{\text{bc}} = \{u \in \mathcal{F}(\Omega) \mid \alpha u + \beta \nabla u \cdot \mathbf{n}|_\Gamma = g\}$ that intersect the PDE zero-residual solution manifold $\mathcal{S}_{\text{res}}$. The well-posedness of the resulting boundary value problem depends on whether these constraints are sufficient to eliminate the intrinsic invariances of the differential operator. We have visualized this in Figure 2.

**Unique Solution.** When the zeroth-order term is present (i.e., $\alpha > 0$, encompassing Dirichlet and Robin conditions) and standard regularity and coercivity assumptions hold, the constraint removes the additive invariance associated with the PDE operator. Consequently, the intersection $\mathcal{S}_{\text{bc}} \cap \mathcal{S}_{\text{res}}$ reduces to a singleton, yielding a unique solution.

**Non-Unique Solution.** When only the derivative term is present (i.e., $\alpha = 0$, corresponding to the Neumann condition), the constraint does not eliminate the additive constant mode for differential operators depending only on derivatives. As a result, if $u \in \mathcal{S}_{\text{bc}} \cap \mathcal{S}_{\text{res}}$, then $u + c$ also belongs to this intersection for any constant $c \in \mathbb{R}$. Therefore, the solution is not unique unless normalization is imposed.

### 4.2. Optimization Dynamics Analysis

In this section, we identify a typical optimization dynamics of standard PINNs from the perspective of gradient competition, as in Figure 3. Note that we do not claim such a decomposition occurs in all possible PINN optimization scenarios. However, for a PDE with large coefficients, it constitutes a representative behavioral pattern, which we demonstrate in Appendix H by an experiment.

### 4.2.1. PHASE I: RAPID DESCENT INTO LOSS VALLEY

As observed by Wang et al. (2021), the gradient magnitude of the PDE residual term typically dominates that of the

(a) *Phase I*      (b) *Phase II* (Success)      (c) *Phase II* (Failure)

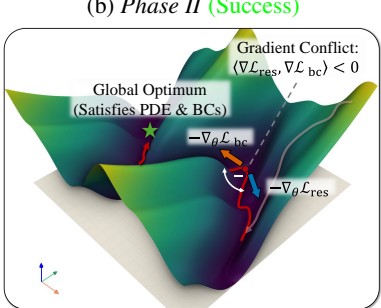 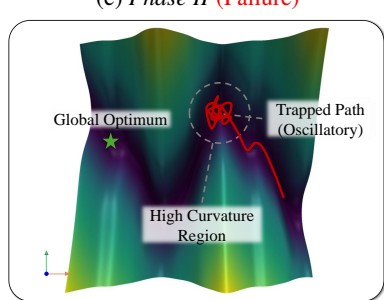

Dominated by $\|\nabla_\theta \mathcal{L}_{\text{res}}\| \gg \|\nabla_\theta \mathcal{L}_{\text{bc}}\|$, reaching $\mathcal{L}_{\text{res}} \leq \varepsilon$.    Guided by $\mathcal{L}_{\text{bc}}$ along the valley, slowed by conflict.    High curvature causes oscillatory updates, trapping the parameter in sub-optimal local minima.

*Figure 3.* Schematic illustration of the two-phase optimization dynamics of PINNs under gradient conflict in **parameter space**. (a) *Phase I*: dominance of the PDE residual gradient leads to a rapid descent into the low-residual loss valley. (b) *Phase II* (success): boundary-condition gradients steer the optimizer along the valley toward the global optimum. (c) *Phase II* (failure): high-curvature regions induce oscillatory updates, trapping the parameter in suboptimal minima.

boundary condition term in a PDE with large coefficients, namely $\|\nabla_\theta \mathcal{L}_{\text{res}}\| \gg \|\nabla_\theta \mathcal{L}_{\text{bc}}\|$, especially during the early stages of training. Consequently, the optimization dynamics is initially governed by the PDE residual. Under this regime, gradient descent follows

$$\theta_{t+1} \approx \theta_t - \eta \nabla_\theta \mathcal{L}_{\text{res}}(\theta_t), \qquad (7)$$

which drives the model to rapidly fall into the low-loss set $\mathcal{M}_{\text{res}} = \{\theta \mid \mathcal{L}_{\text{res}}(\theta) \leq \varepsilon\}$ corresponding to the loss valley induced by the PDE residual, where the threshold $\varepsilon > 0$ marks the transition from residual-dominated to boundary-influenced gradients. Since this valley is generally high-dimensional, while the current landing point is determined by both model initialization and loss landscape, the current output value may deviate substantially from the global optimum that satisfies the boundary conditions.

### 4.2.2. PHASE II: MEANDERING WITHIN THE VALLEY

In the subsequent training phase, the model is confined within the low-loss valley $\mathcal{M}_{\text{res}}$ by the PDE residual gradient $\nabla_\theta \mathcal{L}_{\text{res}}$, while the boundary condition gradient $\nabla_\theta \mathcal{L}_{\text{bc}}$ guides the parameters directly toward the global optimum. Therefore, we establish the following lemma:

**Lemma 4.2** (Gradient Conflict from Perspective of Loss Valley). *Near the PDE loss valley $\mathcal{M}_{\text{res}}$, the residual gradient $\nabla_\theta \mathcal{L}_{\text{res}}$ is normal to this manifold, whereas the boundary condition gradient $\nabla_\theta \mathcal{L}_{\text{bc}}$ generically contains a normal component pointing in the opposite direction. Then the occurrence of a negative inner product*

$$\langle \nabla_\theta \mathcal{L}_{\text{res}}(\theta), \nabla_\theta \mathcal{L}_{\text{bc}}(\theta) \rangle < 0 \qquad (8)$$

*occurs generically during the training process. This leads to a structural gradient conflict during PINN optimization.*

The proof of this lemma is provided in Appendix B. As a result of this conflict, the model parameter is forced to

follow a tortuous trajectory along the valley floor rather than a direct path toward the global optimum, leading to a substantial degradation in convergence speed.

A more severe difficulty arises when the PDE residual-induced loss valley exhibits high curvature along the optimization trajectory. In such cases, the gradient signal provided by the boundary conditions is insufficient to overcome the local geometric constraints from the PDE residual, leading to oscillatory and stagnant update directions. These pathological oscillations hinder the smooth traversal of the valley floor and often trap the parameters in sub-optimal local minima, ultimately slowing or preventing convergence to the true solution.

## 5. Methodology

Based on the above analysis, we redesign the PINN loss function to minimize the extent of optimization exploration during *Phase II*. By expanding the range of feasible solutions and improving the position of entering the valley, we significantly reduce the likelihood of entering regions with high curvature that hinder convergence.

### 5.1. Aligned Constraints

We consider a general steady state PDE given by Equation (1). At the interior collocation points $\{x_i\}_{i=1}^{N_{\text{res}}} \subset \Omega$, its discrete form can be written as

$$\mathcal{N}[u_\theta](x_i) = \underbrace{\mathcal{Z}[u_\theta](x_i)}_{\text{zeroth-order term}} + \underbrace{\mathcal{D}[u_\theta](x_i)}_{\text{derivative terms}} = \underbrace{f(x_i)}_{\text{source term}}. \quad (9)$$

Similarly, boundary conditions imposed at $\{x_b\}_{b=1}^{N_{\text{bc}}} \subset \Gamma$ are expressed in a unified form as

$$\mathcal{B}[u_\theta](x_b) = \underbrace{\alpha_b\, u_\theta(x_b)}_{\text{zeroth-order term}} + \underbrace{\beta_b\, \nabla u_\theta(x_b)\cdot \mathbf{n}_b}_{\text{normal derivative term}} = \underbrace{g(x_b)}_{\text{source term}}.$$
$$(10)$$

This formulation encompasses Dirichlet ($\beta_b = 0$), Neumann ($\alpha_b = 0$), and Robin ($\alpha_b, \beta_b > 0$) as special cases.

For a well-posed problem, a PDE and boundary condition with a nonzero zeroth-order term can remove the additive constant ambiguity. However, it also constrains the solution to a fixed point. As discussed in Section 4.2, even when the model is already initialized close to the global optimum, the PDE residual term may still steer the parameter into a valley far from the true solution, thus significantly slowing convergence. To address this issue, a possible approach is to relax the constraint range, thereby making it easier for the optimizer to identify a global optimum in the loss valley that simultaneously satisfies PDE and boundary conditions.

Therefore, we introduce a solvable additive constant offset $c \in \mathbb{R}$ for all zeroth-order terms into the loss formulation, thus transforming the task into a controllable ill-posed problem. By explicitly solving (linear) or performing a small number of iterations (nonlinear) within each backpropagation step, the optimal value of $c$ is determined, which permits a controlled translation along the additive-invariant direction of the PDE solution manifold during the entire training process. Specifically, we define interior and boundary residuals as

$$
\begin{aligned}
r_i &:= \mathcal{Z}[u_\theta](x_i) + \mathcal{D}[u_\theta](x_i) - f(x_i), \\
s_b &:= \alpha_b\, u_\theta + \beta_b\, \nabla u_\theta \cdot \mathbf{n}_b - g(x_b),
\end{aligned} \tag{11}
$$

then their aligned constraint form can be defined as

$$
\begin{aligned}
\bar{r}_i &:= \mathcal{Z}[u_\theta + c](x_i) + \mathcal{D}[u_\theta](x_i) - f(x_i), \\
\bar{s}_b &:= \alpha_b\, (u_\theta + c) + \beta_b\, \nabla u_\theta \cdot \mathbf{n}_b - g(x_b).
\end{aligned} \tag{12}
$$

Accordingly, the aligned PDE residual loss and boundary condition loss are given by

$$
\mathcal{L}_{\text{res}}^{\text{alg}} = \frac{1}{N_{\text{res}}} \sum_{i=1}^{N_{\text{res}}} |\bar{r}_i|^2, \quad \mathcal{L}_{\text{bc}}^{\text{alg}} = \frac{1}{N_{\text{bc}}} \sum_{b=1}^{N_{\text{bc}}} |\bar{s}_b|^2. \tag{13}
$$

The optimal offset $c$ is then obtained by solving a one-dimensional auxiliary sub-optimization problem

$$
\arg\min_{c \in \mathbb{R}} \; w_{\text{res}} \mathcal{L}_{\text{res}}^{\text{alg}}(c) + w_{\text{bc}} \mathcal{L}_{\text{bc}}^{\text{alg}}(c). \tag{14}
$$

**Linear Case.** For linear zeroth-order terms, this problem admits a closed-form solution:

$$
c = -\frac{\frac{w_{\text{res}}}{N_{\text{res}}} \sum_{i=1}^{N_{\text{res}}} \gamma_i r_i + \frac{w_{\text{bc}}}{N_{\text{bc}}} \sum_{b=1}^{N_{\text{bc}}} \alpha_b s_b}{\frac{w_{\text{res}}}{N_{\text{res}}} \sum_{i=1}^{N_{\text{res}}} \gamma_i^2 + \frac{w_{\text{bc}}}{N_{\text{bc}}} \sum_{b=1}^{N_{\text{bc}}} \alpha_b^2}, \tag{15}
$$

where $\gamma_i$ denotes the coefficient of the zeroth-order terms in PDE, i.e., $\mathcal{Z}[u](x_i) = \gamma_i u(x_i)$. We derived this in Appendix D.1. In practice, $u_\theta$ is explicitly detached from $c$ to prevent gradient propagation through this offset.

**Nonlinear Case.** For nonlinear zeroth-order terms, deriving an optimal value for $c$ generally requires iterative solvers.

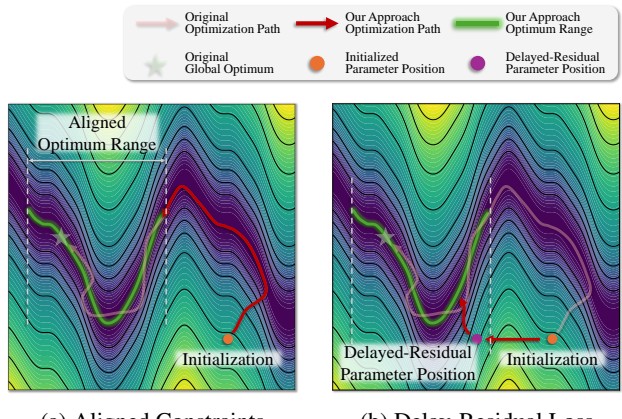

(a) Aligned Constraints  (b) Delay-Residual Loss

*Figure 4.* Conceptual illustration of the proposed optimization mechanism. (a) Aligned constraints enlarge the admissible solution range, while (b) delaying the residual loss guides the optimizer into the PDE loss valley from a favorable region, enabling smoother convergence to the global optimum.

Here, in the early iteration steps, we recommend using Newton's method several times to update $c$. For more complex PDEs, first-order methods such as gradient descent can also be used as alternatives. We provide details in Appendix D.2.

By introducing this additive constant $c$, the network output is no longer constrained to a fixed solution but is instead allowed to translate along this constant direction. Therefore, when training converges,

$$
u^*(x_i) = u_\theta(x_i) + c. \tag{16}
$$

The true solution $u^*$ can be obtained from $u_\theta$ by explicitly eliminating the learned offset $c$.

From an optimization standpoint, this aligned constraint formulation substantially enlarges the overlap between the set of boundary-admissible solutions and the PDE-admissible solution manifold. As a result, even if the parameter falls into a loss valley that is far from the original global optimum at *Phase I*, the optimizer can still efficiently locate an additive shift that is compatible with both constraints within a higher-dimensional expansion. We rigorously prove and discuss this enlargement in Appendix C, and provide the full pipeline in Algorithm 1 in Appendix E.

### 5.2. Delay Factor for Residual Loss

From the perspective of optimization dynamics, another way to shorten the optimization path is to directly bypass the high-curvature regions of the PDE loss landscape. Consequently, we define a time-dependent gating function $\lambda(t) \in [0, 1]$ to guide the parameters to rapidly approach the affine subspace that satisfies the boundary conditions at the early stage of training. The total loss can be written as

$$
\mathcal{L}(\theta; t) = w_{\text{res}}\, \lambda(t)\, \mathcal{L}_{\text{res}}^{\text{alg}} + w_{\text{bc}}\, \mathcal{L}_{\text{bc}}^{\text{alg}}. \tag{17}
$$

*Table 2.* Experimental results on different PDEs with various backbone architectures. Stp: the first iteration step at which relative $L_2$ falls below the threshold (lower is better); $L_2$: mean relative $L_2$ error with standard deviation at a fixed iteration $T_{\min}$ (lower is better); The best results are highlighted in **bold**, and the second-best results are underlined. If the performance of any one of the five seeds does not reach the threshold within the fixed iteration budget $T_{\max}$, it is treated as unsuccessful (reported as '−').

| | | MLP | | | | PirateNets | | | | PINNsFormer | | | |
|---|---|---|---|---|---|---|---|---|---|---|---|---|---|
| | | Heat | Pois. | NS | Helm. | Heat | Pois. | NS | Helm. | Heat | Pois. | NS | Helm. |
| PINN (2019) | Stp | $10127_{\pm2839}$ | — | $8005_{\pm1191}$ | $15375_{\pm3583}$ | $\underline{1141}_{\pm11}$ | — | $8678_{\pm5523}$ | $3875_{\pm2445}$ | $1599_{\pm99}$ | $3197_{\pm296}$ | $389_{\pm17}$ | — |
| | $L_2$ | $7.52\text{e-}3_{\pm4\text{e-}3}$ | — | $1.11\text{e-}2_{\pm5\text{e-}3}$ | $2.21\text{e-}2_{\pm1\text{e-}2}$ | $\underline{8.56\text{e-}3}_{\pm5\text{e-}3}$ | — | $8.48\text{e-}2_{\pm6\text{e-}2}$ | $7.29\text{e-}3_{\pm1\text{e-}2}$ | $3.40\text{e-}2_{\pm3\text{e-}2}$ | $2.86\text{e-}1_{\pm7\text{e-}2}$ | $4.49\text{e-}3_{\pm1\text{e-}3}$ | — |
| $L^\infty$-PINN (2022a) | Stp | — | $12648_{\pm6212}$ | $4768_{\pm552}$ | $\underline{2801}_{\pm205}$ | $7721_{\pm2881}$ | — | $\underline{5545}_{\pm5413}$ | $\underline{1501}_{\pm595}$ | $1912_{\pm290}$ | $3142_{\pm231}$ | $415_{\pm97}$ | $\underline{3435}_{\pm1058}$ |
| | $L_2$ | — | $5.07\text{e-}1_{\pm2\text{e-}1}$ | $8.02\text{e-}3_{\pm3\text{e-}3}$ | $\underline{1.34\text{e-}3}_{\pm4\text{e-}4}$ | $1.01\text{e-}1_{\pm9\text{e-}2}$ | — | $9.12\text{e-}2_{\pm7\text{e-}2}$ | $4.92\text{e-}3_{\pm4\text{e-}3}$ | $1.41\text{e-}1_{\pm1\text{e-}1}$ | $2.00\text{e-}1_{\pm2\text{e-}2}$ | $1.38\text{e-}2_{\pm1\text{e-}2}$ | $\underline{1.97\text{e-}3}_{\pm5\text{e-}4}$ |
| SA-PINN (2023) | Stp | $8958_{\pm2501}$ | $8355_{\pm3867}$ | $6888_{\pm2902}$ | $8959_{\pm2507}$ | $4713_{\pm1043}$ | $9501_{\pm3755}$ | $8930_{\pm5914}$ | $4598_{\pm3340}$ | $1649_{\pm227}$ | $\mathbf{1888}_{\pm118}$ | $\underline{330}_{\pm9}$ | — |
| | $L_2$ | $\underline{4.71\text{e-}3}_{\pm2\text{e-}3}$ | $1.95\text{e-}1_{\pm2\text{e-}1}$ | $9.79\text{e-}3_{\pm4\text{e-}3}$ | $8.07\text{e-}3_{\pm5\text{e-}3}$ | $8.64\text{e-}2_{\pm1\text{e-}1}$ | $6.13\text{e-}1_{\pm5\text{e-}1}$ | $8.71\text{e-}2_{\pm6\text{e-}2}$ | $6.44\text{e-}3_{\pm4\text{e-}3}$ | $1.23\text{e-}2_{\pm1\text{e-}3}$ | $\mathbf{4.41\text{e-}2}_{\pm1\text{e-}2}$ | $6.76\text{e-}3_{\pm5\text{e-}3}$ | — |
| BRDR (2025) | Stp | $15213_{\pm2175}$ | $\underline{6137}_{\pm795}$ | $5462_{\pm605}$ | $13131_{\pm2505}$ | $5245_{\pm1058}$ | $\underline{8276}_{\pm1326}$ | $9427_{\pm4083}$ | $5831_{\pm6922}$ | $1941_{\pm203}$ | $2333_{\pm214}$ | $367_{\pm13}$ | — |
| | $L_2$ | $5.58\text{e-}3_{\pm2\text{e-}3}$ | $\underline{8.51\text{e-}2}_{\pm7\text{e-}2}$ | $8.66\text{e-}3_{\pm3\text{e-}3}$ | $2.05\text{e-}2_{\pm1\text{e-}2}$ | $5.51\text{e-}1_{\pm3\text{e-}1}$ | $\underline{5.20\text{e-}1}_{\pm3\text{e-}1}$ | $9.26\text{e-}2_{\pm6\text{e-}2}$ | $6.81\text{e-}3_{\pm1\text{e-}2}$ | $1.67\text{e-}2_{\pm1\text{e-}2}$ | $8.65\text{e-}2_{\pm4\text{e-}2}$ | $6.25\text{e-}3_{\pm1\text{e-}3}$ | — |
| DB-PINN (2025a) | Stp | $6988_{\pm329}$ | $9451_{\pm6738}$ | $\underline{3981}_{\pm294}$ | $11005_{\pm1662}$ | $12689_{\pm6402}$ | — | $6054_{\pm4412}$ | $2304_{\pm1113}$ | $\underline{1159}_{\pm132}$ | — | $331_{\pm36}$ | $4882_{\pm245}$ |
| | $L_2$ | $6.82\text{e-}2_{\pm1\text{e-}2}$ | $4.56\text{e-}1_{\pm2\text{e-}1}$ | $\underline{7.89\text{e-}3}_{\pm2\text{e-}3}$ | $1.22\text{e-}2_{\pm1\text{e-}2}$ | $3.12\text{e-}1_{\pm2\text{e-}1}$ | — | $\underline{3.47\text{e-}2}_{\pm2\text{e-}2}$ | $3.69\text{e-}3_{\pm2\text{e-}3}$ | $\underline{1.67\text{e-}3}_{\pm8\text{e-}3}$ | — | $5.69\text{e-}3_{\pm2\text{e-}3}$ | $5.37\text{e-}3_{\pm2\text{e-}3}$ |
| **CAML (Ours)** | **Stp** | $\mathbf{1577}_{\pm130}$ | $\mathbf{3868}_{\pm1932}$ | $\mathbf{1269}_{\pm966}$ | $\mathbf{1004}_{\pm450}$ | $\mathbf{713}_{\pm496}$ | $\mathbf{4880}_{\pm726}$ | $\mathbf{2597}_{\pm748}$ | $\mathbf{410}_{\pm401}$ | $\mathbf{746}_{\pm115}$ | $\underline{1936}_{\pm78}$ | $\mathbf{201}_{\pm54}$ | $\mathbf{1659}_{\pm1604}$ |
| | **$L_2$** | $\mathbf{1.16\text{e-}3}_{\pm8\text{e-}5}$ | $\mathbf{5.00\text{e-}3}_{\pm9\text{e-}4}$ | $\mathbf{4.73\text{e-}3}_{\pm3\text{e-}4}$ | $\mathbf{1.56\text{e-}4}_{\pm5\text{e-}5}$ | $\mathbf{4.98\text{e-}3}_{\pm4\text{e-}3}$ | $\mathbf{3.24\text{e-}2}_{\pm1\text{e-}2}$ | $\mathbf{2.79\text{e-}2}_{\pm2\text{e-}2}$ | $\mathbf{1.19\text{e-}3}_{\pm5\text{e-}4}$ | $\underline{4.04\text{e-}3}_{\pm9\text{e-}4}$ | $8.29\text{e-}2_{\pm1\text{e-}2}$ | $\mathbf{4.07\text{e-}3}_{\pm2\text{e-}3}$ | $\mathbf{9.10\text{e-}4}_{\pm3\text{e-}4}$ |

A simple and reproducible choice for $\lambda(t)$ is a piecewise linear schedule:

$$\lambda(t) = \begin{cases} 0, & t < t_d, \\ (t - t_d)/t_r, & t_d \le t < t_d + t_r, \\ 1, & t \ge t_d + t_r, \end{cases} \quad (18)$$

where $t_d$ is the delay step number and $t_r$ is the ramp length.

Although Equation (17) resembles a dynamic reweighting scheme, the proposed delay factor fundamentally alters the optimization dynamics. Fixed or adaptive weighting strategies still allow the residual term to dominate early training, thereby pulling the parameters into an unfavorable location within the residual valley. In contrast, the delay factor explicitly prevents this, thus reducing the probability of entering a valley that is far from the global optimum.

## 6. Experiments and Results

Our experiments aim to answer the following five questions: *RQ1.* How does the proposed approach compare with state-of-the-art PINN loss variants in terms of efficiency, stability, and sensitivity to initialization across different physical problems? *RQ2.* Can the proposed aligned constraint formulation effectively mitigate gradient conflicts between PDE residuals and boundary conditions during PINN training and shorten the optimization path? *RQ3.* What is the contribution of aligned constraints and delay-residual optimization? *RQ4.* How does CAML integrate with the existing conflict-aware optimizers? And *RQ5.* To which scenarios is CAML applicable?

### 6.1. Benchmarks and Experiments Setup

We selected four representative and challenging benchmarks across thermodynamics, fluid mechanics and electromagnetism, including: *(i)* Heat, a heat conduction problem with composite boundary conditions, *(ii)* Poisson, a Poisson problem with complex nonlinear boundary conditions, *(iii)* NS,

a steady-state Navier-Stokes problem with a nonlinear operator, and *(iv)* Helmholtz, a Helmholtz reaction-diffusion problem with complex geometry and high frequency. As baselines, we compare with the following five **loss function-based** PINN variants: *(i)* classic PINN (Raissi et al., 2019), *(ii)* $L^\infty$-PINN (Wang et al., 2022a), *(iii)* SA-PINN (McClenny & Braga-Neto, 2023), *(iv)* BRDR (Chen et al., 2025), and *(v)* DB-PINN (Zhou et al., 2025a).

To systematically evaluate the performance of our method across different parameter spaces, we consider three representative model architectures in all benchmarks: *(i)* an MLP-based PINN, *(ii)* PirateNets (Wang et al., 2024), a physics-informed ResNet, and *(iii)* PINNsFormer (Zhao et al., 2024), a physics-informed Transformer. For all experiments, we keep the network structures and hyperparameter configurations consistent with the original implementations, and only replace the loss function. The optimizer is Adam (Kingma & Ba, 2015). We implemented the algorithms using PyTorch (Paszke et al., 2019) and the experiments are conducted on an NVIDIA RTX 5090 GPU.

In each experiment, we selected five random seeds for initialization. In addition to *(i)* relative $L_2$ error at the same iteration step $T_{\min}$ and *(ii)* the number of iterations required for the relative $L_2$ error to reach a predefined accuracy threshold, we further introduce the following statistic indicator in part of the experiments to evaluate the stability of optimization: *(iii)* full-progress gradient cosine similarity

$$\cos(\phi) = \frac{\langle \nabla_\theta \mathcal{L}_{\text{res}}, \nabla_\theta \mathcal{L}_{\text{bc}} \rangle}{\|\nabla_\theta \mathcal{L}_{\text{res}}\| \|\nabla_\theta \mathcal{L}_{\text{bc}}\|} \quad (19)$$

following previous work (Du et al., 2018; Wang et al., 2021), which characterizes the degree of conflict between the residual and boundary-condition gradients. The higher this value, the more similar the directions of the two gradients will be. We report the fraction of training iterations (over a fixed total number of iterations) for which $\cos(\phi) > 0$. All details of the experimental setup are in Appendix I.

*Table 3.* Full-process gradient cosine similarity $\cos(\phi)$ on Heat and Poisson benchmarks. $\cos(\phi)$ is only a diagnostic statistic, and should not be interpreted as a performance metric.

| | | MLP | | PirateNets | | PINNsFormer | |
|---|---|---|---|---|---|---|---|
| | | Heat | Pois. | Heat | Pois. | Heat | Pois. |
| PINN | (2019) | 5.89% | 3.85% | 1.44% | 4.09% | 5.13% | 7.07% |
| $L^\infty$-PINN | (2022a) | 11.98% | 62.81% | 40.22% | 84.08% | 54.68% | 87.78% |
| SA-PINN | (2023) | 4.95% | 9.73% | 5.72% | 11.18% | 5.27% | 6.98% |
| BRDR | (2025) | 4.57% | 21.28% | 6.56% | 4.63% | 4.02% | 8.00% |
| DB-PINN | (2025a) | 0.02% | 3.19% | 7.82% | 10.01% | 5.17% | 16.32% |
| **CAML** | **(Ours)** | 39.16% | 52.75% | 39.41% | 31.58% | 41.45% | 55.50% |

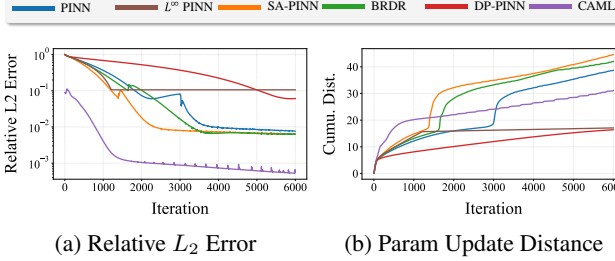

(a) Relative $L_2$ Error        (b) Param Update Distance

*Figure 5.* Relative $L_2$ error and cumulative parameter update distance $\sum \|\theta_{t+1} - \theta_t\|_2$ versus training iterations on Heat benchmark (MLP backbone).

## 6.2. Results and Discussions

**RQ1: Performance Analysis.** Table 2 presents quantitative comparisons across four PDE benchmarks and three backbone architectures. CAML consistently achieves the best or second-best $L_2$ accuracy in nearly all settings, while requiring substantially fewer iterations to reach the target error. In contrast, several baselines exhibit large variance or fail entirely when some seeds do not converge within the iteration budget. CAML is successful in most trials and shows significantly reduced variance in both convergence steps and final error. This robustness aligns with the analysis in Section 4.2: by enlarging the admissible solution set via manifold lifting, CAML mitigates sensitivity to initialization-dependent minima in the residual valley. Moreover, its consistent performance across architectures indicates that the gains arise from improved optimization geometry induced by aligned constraints, rather than architecture-specific effects.

**RQ2: Gradient Conflict Mitigation.** Table 3 reports the fraction of training iterations with positive cosine similarity between residual and boundary gradients. For standard PINNs, this fraction is typically less than 10%, indicating persistent gradient conflict throughout training. In contrast, CAML achieves an order-of-magnitude increase across all backbone architectures, demonstrating its effectiveness in alleviating gradient conflicts. Notably, $L^\infty$-PINN forces gradient alignment at the most critical points. As a result, the probability of gradient consistency can be statistically higher. However, this alignment is local and passive, and does not fundamentally fix the pathological structure of the

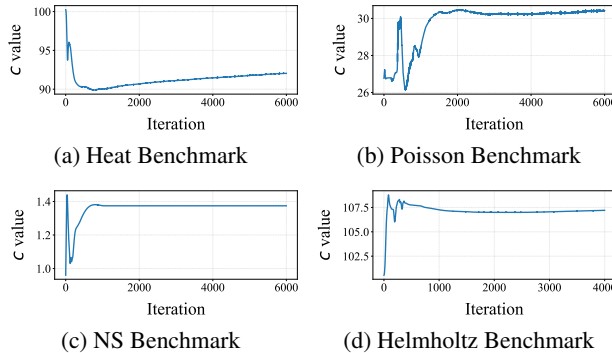

(a) Heat Benchmark        (b) Poisson Benchmark

(c) NS Benchmark        (d) Helmholtz Benchmark

*Figure 6.* Variation of the $c$ curve across iterations on all benchmarks (MLP backbone).

loss landscape. Consequently, it is not equivalent to genuine progress toward the correct solution.

Figure 5 presents the relative error and parameter update distance curves of all loss functions for the MLP model on the Heat benchmark. The results indicate that CAML exhibits a substantially faster optimization speed from initialization compared to other baselines. Moreover, excluding the two suboptimal methods (DB-PINN and $L^\infty$-PINN), CAML traverses a shorter path in the parameter space.

Figure 6 shows the changes in $c$ during the training iterations, which gradually converge to a fixed value as training progresses. In the linear cases, $c$ admits an analytical optimum, and thus the loss is strictly reduced regardless of its value. Therefore, larger variations in $c$ at initial stage are, in fact, expected to yield greater stability, as they indicate that CAML actively aligns more pathological conflicts. In the nonlinear case, however, the small errors introduced by Newton's method can affect the accuracy of the final solution. This is why we introduce $t_c$ and fix $c$ once its variation becomes sufficiently small, as stated in Appendix D.2, thereby avoiding training noise caused by minor fluctuations.

We also analyze computational overhead, the difference between aligned constraints and learnable bias, as well as parameter sensitivity, as detailed in Appendices J, K, and L.

## 6.3. Ablation Study

To evaluate the contribution of each component, we conducted an ablation study under four configurations: *(i)* original PINN, *(ii)* Aligned Constraint (AC) only, *(iii)* Delay-Residual (DR) only, and *(iv)* full CAML. We conducted experiments on MLP on both Heat and Poisson benchmarks, and all configurations are consistent with main experiment.

**RQ3: Ablation Study.** We report all results in Table 4. On the Heat benchmark, AC makes the most significant contribution, markedly improving gradient consistency and accelerating convergence. When DR is applied independently,

*Table 4.* Ablation study on Heat and Poisson benchmarks (MLP backbone). The definition of Stp, $L_2$ and $\cos(\phi)$ is the same as that mentioned in previous experiment. Configurations that cannot achieve the expected accuracy within the maximum training budget $T_{\max}$ are indicated by '−'.

| | Heat | | | Poisson | | |
|---|---|---|---|---|---|---|
| | Stp | $L_2$ | $\cos(\phi)$ | Stp | $L_2$ | $\cos(\phi)$ |
| PINN | $10127_{\pm 2839}$ | $7.52\text{e-}3_{\pm 4\text{e-}3}$ | 5.89% | − | − | 3.85% |
| AC-Only | $1491_{\pm 186}$ | $1.15\text{e-}3_{\pm 6\text{e-}5}$ | 36.12% | − | − | 45.22% |
| DR-Only | $8367_{\pm 991}$ | $7.16\text{e-}3_{\pm 4\text{e-}3}$ | 4.89% | $9648_{\pm 3075}$ | $4.63\text{e-}2_{\pm 3\text{e-}2}$ | 14.62% |
| CAML | $1577_{\pm 130}$ | $1.16\text{e-}3_{\pm 8\text{e-}5}$ | 39.16% | $3868_{\pm 1932}$ | $5.00\text{e-}3_{\pm 9\text{e-}4}$ | 52.75% |

*Table 5.* Comparison of CAML with different optimizers on Heat benchmarks (MLP backbone). The definitions of Stp, $L_2$, and $\cos(\phi)$ are the same as those mentioned in the previous experiment. We recorded the values of $L_2$ at both $T_{\min}$ and $T_{\max}$.

| | | Stp | $L_2@T_{\min}$ | $L_2@T_{\max}$ | $\cos(\phi)$ |
|---|---|---|---|---|---|
| Adam | (2015) | $1577_{\pm 130}$ | $1.16\text{e-}3_{\pm 8\text{e-}5}$ | $1.12\text{e-}3_{\pm 4\text{e-}5}$ | 39.16% |
| L-BFGS | (1989) | $55_{\pm 3}$ | $6.93\text{e-}4_{\pm 3\text{e-}6}$ | $6.93\text{e-}4_{\pm 2\text{e-}6}$ | 26.00% |
| DCGD | (2024) | $991_{\pm 372}$ | $1.15\text{e-}3_{\pm 8\text{e-}5}$ | $1.08\text{e-}3_{\pm 3\text{e-}5}$ | 36.37% |
| ConFIG | (2025) | $738_{\pm 238}$ | $1.13\text{e-}3_{\pm 6\text{e-}5}$ | $1.06\text{e-}3_{\pm 2\text{e-}5}$ | 37.14% |

it also yields partial improvements in convergence speed. However, in the full CAML setting, no additional benefit is observed beyond that provided by AC alone. In contrast, for benchmarks such as Poisson with strongly nonlinear boundary conditions, although AC substantially enhances gradient consistency, it still fails to reach the target accuracy within the fixed epoch budget. DR, on the other hand, significantly improves convergence efficiency, and its combination with AC achieves the best overall performance. These results suggest that for problems with relatively simple boundary conditions, AC alone is sufficient to substantially enhance training stability, whereas for more complex boundary conditions, the integration of both mechanisms yields superior performance.

### 6.4. Optimizer Selection

In this section, we compare CAML's performance across four optimizers and verify its compatibility with a range of advanced optimizers. The optimizers we have chosen include: *(i)* Adam (Kingma & Ba, 2015), learning rate $\eta = 1\text{e-}3$, $\beta_1 = 0.9$, $\beta_2 = 0.999$, *(ii)* L-BFGS (Liu & Nocedal, 1989), we use PyTorch L-BFGS with strong Wolfe line search (default), max history size $m = 50$, max line-search steps 25, *(iii)* DCGD (Hwang & Lim, 2024), same settings as Adam, and *(iv)* ConFIG (Liu et al., 2025), same settings as Adam. We used the Heat benchmark with the MLP backbone, following the same setup as the main experiments. Because L-BFGS steps are expensive, we limit the number of L-BFGS iterations to $T_{\min}^{\text{LBFGS}} = 300$ (which typically matches or exceeds the wall-clock time of 6000 Adam steps) and $T_{\max}^{\text{LBFGS}} = 1000$. In addition, the delayed residuals were disabled in L-BFGS to prevent the gradient explosion caused by the distortion of the Hessian approximation. We also compared the final precision of all optimizers, and Table 5 provides a compact summary.

**RQ4: Optimizer Selection.** We observe that conflict-aware methods (DCGD and ConFIG) consistently provide more favorable optimization dynamics than standard first- and second-order baselines. ConFIG achieves the best trade-off between convergence speed and final accuracy, indicating that explicitly resolving gradient conflicts benefits both ef-

ficiency and stability. In addition, L-BFGS achieves the best final accuracy among all optimizers, indicating that second-order curvature information is particularly effective in refining the solution once the optimization trajectory enters a favorable region. Based on these observations, we recommend a two-stage training strategy: first, use a conflict-aware first-order optimizer for rapid convergence, and then switch to L-BFGS for fine-grained refinement.

### 6.5. Applicability and Failure Discussion

**RQ5: Applicability and Failure Modes.** CAML targets the residual-induced loss valley analyzed in Section 4, and is most effective when *(i)* the PDE or BC contains at least one non-trivial zeroth-order term, *(ii)* PDE coefficients are large enough to induce $\|\nabla_\theta \mathcal{L}_{\text{res}}\| \gg \|\nabla_\theta \mathcal{L}_{\text{bc}}\|$, or *(iii)* BCs are composite or strongly nonlinear. Conversely, CAML degenerates to standard PINN under pure Neumann BCs without zeroth-order terms (the offset is annihilated by derivative operators), and yields limited gains when the landscape is already well-conditioned ($c \approx 0$ throughout training, see our demonstration in Appendix M).

## 7. Conclusion

This paper explains the gradient pathologies of PINNs under composite boundary conditions from the perspective of loss geometry, showing that gradient conflicts between PDE residuals and boundary terms hinder optimization. To address this, we propose CAML, which aligns constraints via manifold lifting and introduces a residual-delay strategy to stabilize training. Extensive experiments demonstrate that CAML consistently improves convergence and stability across diverse PDEs and network architectures, highlighting its potential as a general PINN loss framework.

Future work will extend this approach to time-dependent (dynamic) PDEs with initial conditions. This will require extending our analytical framework to address the more complex three-way gradient interaction and reformulating the optimal offset as a time-dependent function $c(t)$. To achieve this, $c(t)$ could be solved independently at each time step for linear equations, or approximated via time-domain partitioning and low-dimensional parameterizations for nonlinear cases. See Appendix F for more future work.

## Acknowledgements

Yichen Luo is sponsored by the China Scholarship Council for his PhD study at KTH Royal Institute of Technology, Sweden (No. 202408320060). This research was also supported in part by the National Natural Science Foundation of China (Grant No. 72401232) and XJTLU Technical Service (2024-015).

## Impact Statement

This paper presents a methodological effort aimed at improving Physics-Informed Neural Networks (PINNs) by mitigating their inherent gradient pathologies. PINNs are increasingly critical in scientific computing, with broad applications ranging from fluid dynamics and climate modeling to advanced manufacturing and thermal engineering. By significantly improving the convergence, efficiency, and reliability of these models, our proposed CAML framework can accelerate complex engineering designs and scientific discoveries. As a fundamental algorithmic and theoretical contribution, this work does not present any immediate ethical concerns or negative societal consequences.

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

# Part I

# Theoretical Proof

## A. Existence of Loss Valleys in PDE Residual Term

In this section, we provide a rigorous justification for the existence of loss valleys in PINNs. The analysis proceeds in three stages: *(i)* we establish the non-uniqueness of solutions in the functional space induced by the governing differential operator, *(ii)* we show that this functional non-uniqueness implies the existence of flat, connected sets of global minima of the PINN loss, and *(iii)* we demonstrate how such functional non-uniqueness can be mapped to the parameter space of a neural network, thereby forming a loss valley.

### A.1. Non-Uniqueness of Solutions

We first formalize the fact that, in the absence of sufficient boundary or initial conditions, the governing differential operator admits a non-trivial solution set.

**Theorem A.1** (Non-Triviality of the Solution Space). *Let $\mathcal{N}$ be a differential operator that defines the PDE constraint $\mathcal{N}[u] = f$ on a function space $\mathcal{F}(\Omega)$. If $\mathcal{N}$ does not uniquely determine a solution in $\mathcal{F}(\Omega)$ (e.g., due to missing boundary or initial conditions), then the solution set*

$$\mathcal{S}_{\mathrm{res}} := \{u \in \mathcal{F}(\Omega) \mid \mathcal{N}[u] = f\} \tag{20}$$

*is non-trivial and contains non-isolated solutions. Moreover, around any regular solution $u^* \in \mathcal{S}_{\mathrm{res}}$, the solution set locally contains a continuously parameterized family of solutions of dimension at least one.*

*Proof.* Let $u^*$ be a solution that satisfies $\mathcal{N}[u^*] = f$.

(a) Linear case. If $\mathcal{N}$ is linear and the solution is not uniquely determined, then its kernel $\ker(\mathcal{N})$ is non-trivial. For any $v \in \ker(\mathcal{N})$ and scalar $\alpha \in \mathbb{R}$,

$$\mathcal{N}[u^* + \alpha v] = \mathcal{N}[u^*] + \alpha \mathcal{N}[v] = f. \tag{21}$$

Hence, the solution set contains an affine subspace passing through $u^*$ and is therefore non-isolated.

(b) Nonlinear case. For nonlinear operators, the solution set need not be globally finite-dimensional. We therefore restrict attention to a local neighborhood of a regular solution. Assume that $\mathcal{N}$ is Fréchet differentiable in a neighborhood of $u^*$ and that the linearized operator

$$D\mathcal{N}[u^*] : \mathcal{F}(\Omega) \to \mathcal{G}(\Omega) \tag{22}$$

has a non-trivial kernel. Suppose further that $u^*$ is a regular point in the sense that $D\mathcal{N}[u^*]$ satisfies a constant-rank or regular-value condition (e.g., it is surjective and its kernel is complemented in $\mathcal{F}(\Omega)$).

Under these standard assumptions, Banach-space implicit function or constant-rank theorems imply that the zero set of $\mathcal{N}$ is locally a smooth embedded submanifold of $\mathcal{F}(\Omega)$ whose dimension is equal to $\dim \ker(D\mathcal{N}[u^*]) \geq 1$. Consequently, there exists a neighborhood $V$ of $u^*$ and an open set $U \subset \mathbb{R}^k$, $k \geq 1$, such that

$$\mathcal{S}_{\mathrm{res}} \cap V \supset \{u(\cdot; \mathbf{c}) \mid \mathbf{c} \in U\}, \tag{23}$$

where $\mathbf{c}$ parameterizes a continuous family of solutions. This establishes the local non-uniqueness of the solution space. $\square$

### A.2. Functional Non-Uniqueness and Flat Minima

We next show that the functional non-uniqueness of solutions induces flat, connected sets of global minima of the PINN loss in function space.

**Proposition A.2** (Flat Directions in Function Space). *Any continuously parameterized family of solutions*

$$\{u(\cdot; \mathbf{c}) \mid \mathbf{c} \in U \subset \mathbb{R}^k\} \subset \mathcal{S}_{\mathrm{res}} \tag{24}$$

*forms a flat, connected set of global minimizers of loss $\mathcal{L}_{\mathrm{func}}$ in function space.*

*Proof.* By Theorem A.1, the global minimizers of $\mathcal{L}_{\text{func}}$ coincide with $\mathcal{S}_{\text{res}} = \{u \in \mathcal{F}(\Omega) \mid \mathcal{N}[u] = f\}$.

(a) Global minimality. For any $\mathbf{c} \in U$, we have $u(\cdot; \mathbf{c}) \in \mathcal{S}_{\text{res}}$, hence $\mathcal{N}[u(\cdot; \mathbf{c})] = f$. By assumption,

$$\mathcal{L}_{\text{func}}(u(\cdot; \mathbf{c})) = 0. \tag{25}$$

Since $\mathcal{L}_{\text{func}}(u) \geq 0$ for all $u$, the global minimum value is 0, and every element of the family is therefore a global minimizer.

(b) Connectedness. The map $\mathbf{c} \mapsto u(\cdot; \mathbf{c})$ is continuous by Equation 6 in the main text. If $U \subset \mathbb{R}^k$ is connected, then its continuous image $\{u(\cdot; \mathbf{c}) \mid \mathbf{c} \in U\}$ is also connected.

(c) Flatness. For any smooth curve $\mathbf{c}(t) \subset U$, define $u_t := u(\cdot; \mathbf{c}(t))$. From the first part,

$$\mathcal{L}_{\text{func}}(u_t) \equiv 0 \quad \text{for all } t. \tag{26}$$

Hence the loss remains constant along any such direction, so all tangent directions to the solution family are flat directions. Therefore, the continuously parameterized solution family forms a flat, connected set of global minimizers of $\mathcal{L}_{\text{func}}$. $\square$

### A.3. Mapping Functional Valleys to Parameter Space

Finally, we show that functional non-uniqueness can be transferred to the parameter space of a neural network, leading to loss valleys in practice.

Let $u_\theta \in \mathcal{F}(\Omega)$ denote the neural network approximation parameterized by $\theta \in \mathbb{R}^P$, and define the parameter-space loss

$$\mathcal{L}(\theta) := \mathcal{L}_{\text{func}}(u_\theta). \tag{27}$$

**Assumption A.3** (Representability of a Solution Family)**.** There exists a $\mathcal{C}^1$ mapping $\phi : U \subset \mathbb{R}^k \to \mathbb{R}^P$ such that

$$u_{\phi(\mathbf{c})} = u(\cdot; \mathbf{c}) \quad \text{for all } \mathbf{c} \in U, \tag{28}$$

where $\{u(\cdot; \mathbf{c})\}$ is a solution family as in Proposition A.2. Moreover, $\phi$ is locally non-degenerate in the sense that there exists $\mathbf{c}^* \in U$ and a direction $v \in \mathbb{R}^k$ such that

$$J\phi(\mathbf{c}^*) v \neq 0, \tag{29}$$

where $J\phi(\mathbf{c}^*)$ denotes the Jacobian of $\phi$ at $\mathbf{c}^*$.

**Assumption A.4** (Local $\mathcal{C}^2$ Regularity of the Loss)**.** The parameter-space loss $\mathcal{L}(\theta)$ is twice continuously differentiable in a neighborhood of $\Theta_{\text{valley}}$.

Assumption A.3 only requires that the neural network can locally represent a non-isolated solution family, which is consistent with the expressive capacity and over-parameterization of modern neural networks. Assumption A.4 is also standard, since PINNs typically adopt globally differentiable activation functions, such as Tanh, Sigmoid, or Sine, together with differentiable PDE operators. Therefore, the resulting residual loss is generally smooth in the relevant parameter region.

**Theorem A.5** (Existence of Loss Valleys in Parameter Space)**.** *Under Assumptions A.3 and A.4, the parameter-space loss $\mathcal{L}(\theta)$ admits a flat, connected set of global minimizers*

$$\Theta_{\text{valley}} := \{\phi(\mathbf{c}) \mid \mathbf{c} \in U\}. \tag{30}$$

*Moreover, at any $\theta^* \in \Theta_{\text{valley}}$, the Hessian $\nabla_\theta^2 \mathcal{L}(\theta^*)$ has at least one zero eigenvalue corresponding to a flat direction.*

*Proof.* For any $\mathbf{c} \in U$,

$$\mathcal{L}(\phi(\mathbf{c})) = \mathcal{L}_{\text{func}}(u(\cdot; \mathbf{c})) = 0, \tag{31}$$

so $\Theta_{\text{valley}}$ consists entirely of global minimizers. Continuity of $\phi$ implies that $\Theta_{\text{valley}}$ is path-connected.

Fix any $\theta^* = \phi(\mathbf{c}^*) \in \Theta_{\text{valley}}$. Since $\theta^*$ is a global minimizer and $\mathcal{L} \geq 0$, we have

$$\nabla_\theta \mathcal{L}(\theta^*) = 0. \tag{32}$$

By Assumption A.3, choose $v \in \mathbb{R}^k$ such that $w := J\phi(\mathbf{c}^*)v \neq 0$. Let $\mathbf{c}(t) = \mathbf{c}^* + tv$ (for $t$ small) and define $\theta(t) := \phi(\mathbf{c}(t))$, so that $\theta(t) \subset \Theta_{\text{valley}}$ and $\dot{\theta}(0) = w \neq 0$.

Then $\mathcal{L}(\theta(t)) \equiv 0$ for all sufficiently small $t$, hence

$$\frac{d}{dt}\mathcal{L}(\theta(t))\Big|_{t=0} = \nabla_\theta \mathcal{L}(\theta^*)^\top \dot{\theta}(0) = 0, \tag{33}$$

and by Assumption A.4,

$$0 = \frac{d^2 \mathcal{L}}{dt^2}\Big|_{t=0} = \dot{\theta}(0)^\top \nabla_\theta^2 \mathcal{L}(\theta^*) \dot{\theta}(0) = w^\top \nabla_\theta^2 \mathcal{L}(\theta^*) w. \tag{34}$$

Since $\theta^*$ is a (global) minimizer and $\mathcal{L}$ is $\mathcal{C}^2$ near $\theta^*$, the Hessian $\nabla_\theta^2 \mathcal{L}(\theta^*)$ is positive semidefinite. Therefore, the existence of a nonzero $w$ such that $w^\top \nabla_\theta^2 \mathcal{L}(\theta^*) w = 0$ implies that $\nabla_\theta^2 \mathcal{L}(\theta^*)$ has at least one zero eigenvalue. $\qquad\square$

*Remark* A.6 (Inevitability of Loss Valley). The above analysis shows that loss valleys in the PDE residual term arise naturally from the structural non-uniqueness of the underlying PDE when insufficient constraints are imposed. Importantly, the argument relies only on local and generic properties of the solution set and does not require global uniqueness or finite-dimensionality of the full solution manifold. This explains why flat loss landscapes are commonly observed in practical PINN training.

# B. Structural Origin of Gradient Conflict in PINNs

In this section, we provide a theoretical explanation for why gradients induced by PDE residual and boundary condition losses tend to be conflicting during PINN optimization. The analysis proceeds in two steps: *(i)* we characterize the residual-induced low-loss set as a manifold in parameter space, and *(ii)* we show that boundary-driven descent directions generically contain a normal component opposing the residual gradient, yielding a negative inner product.

## B.1. Geometric Decomposition near the Low-Residual Manifold

Let $\mathcal{L}_{\text{res}}(\theta)$ and $\mathcal{L}_{\text{bc}}(\theta)$ denote the PDE residual loss and boundary condition loss, respectively. We define the low-residual set

$$\mathcal{M}_{\text{res}} = \left\{ \theta \in \mathbb{R}^P \mid \mathcal{L}_{\text{res}}(\theta) \leq \varepsilon \text{ such that } \|\nabla_\theta \mathcal{L}_{\text{res}}(\theta)\| < \|\nabla_\theta \mathcal{L}_{\text{bc}}(\theta)\| \right\}, \tag{35}$$

where $\varepsilon > 0$ is a threshold that marks the transition from residual-dominated to boundary-influenced gradients. This set corresponds to the residual-induced loss valley discussed in Appendix A.

**Assumption B.1** (Smooth Manifold Approximation). There exists a neighborhood $\mathcal{U}$ of $\mathcal{M}_{\text{res}}$ such that $\mathcal{M}_{\text{res}} \cap \mathcal{U}$ is a smooth embedded submanifold of $\mathbb{R}^P$. Moreover, for every $\theta \in \mathcal{M}_{\text{res}} \cap \mathcal{U}$, the gradient $\nabla_\theta \mathcal{L}_{\text{res}}(\theta)$ is normal to $\mathcal{M}_{\text{res}}$.

The validity of this assumption is supported from two aspects. Theoretically, Theorem A.1 proves that the zero-residual set $\mathcal{S}_{\text{res}}$ is indeed a smooth manifold in function space, and the smooth parameterization induced by overparameterized neural networks preserves this local structure. Empirically, the extremely large condition numbers of the Hessian and the presence of clear low-curvature subspaces reported in Table 1 also support the existence of such a manifold structure.

**Lemma B.2** (Normal–Tangent Decomposition of Boundary Gradients). *Under Assumption B.1, for any $\theta \in \mathcal{M}_{\text{res}} \cap \mathcal{U}$, the boundary gradient admits an orthogonal decomposition into tangent and normal components with respect to $\mathcal{M}_{\text{res}}$,*

$$\nabla_\theta \mathcal{L}_{\text{bc}}(\theta) = \nabla_\theta^\top \mathcal{L}_{\text{bc}}(\theta) + \nabla_\theta^\perp \mathcal{L}_{\text{bc}}(\theta), \tag{36}$$

*where $\nabla_\theta^\top \mathcal{L}_{\text{bc}}(\theta) \in T_\theta \mathcal{M}_{\text{res}}$ and $\nabla_\theta^\perp \mathcal{L}_{\text{bc}}(\theta) \in N_\theta \mathcal{M}_{\text{res}}$. Consequently,*

$$\langle \nabla_\theta \mathcal{L}_{\text{res}}(\theta), \nabla_\theta \mathcal{L}_{\text{bc}}(\theta) \rangle = \langle \nabla_\theta \mathcal{L}_{\text{res}}(\theta), \nabla_\theta^\perp \mathcal{L}_{\text{bc}}(\theta) \rangle. \tag{37}$$

*Proof.* By definition of orthogonal projection, we have

$$\nabla_\theta \mathcal{L}_{\text{bc}}(\theta) = P_T(\theta)\nabla_\theta \mathcal{L}_{\text{bc}}(\theta) + P_N(\theta)\nabla_\theta \mathcal{L}_{\text{bc}}(\theta), \tag{38}$$

with $P_T(\theta)\nabla_\theta \mathcal{L}_{\text{bc}}(\theta) \in T_\theta \mathcal{M}_{\text{res}}$ and $P_N(\theta)\nabla_\theta \mathcal{L}_{\text{bc}}(\theta) \in N_\theta \mathcal{M}_{\text{res}}$. Under Assumption B.1, $\nabla_\theta \mathcal{L}_{\text{res}}(\theta) \in N_\theta \mathcal{M}_{\text{res}}$, hence it is orthogonal to $T_\theta \mathcal{M}_{\text{res}}$. Therefore,

$$\langle \nabla_\theta \mathcal{L}_{\text{res}}(\theta), P_T(\theta)\nabla_\theta \mathcal{L}_{\text{bc}}(\theta) \rangle = 0, \tag{39}$$

and the stated identity follows. $\qquad\square$

## B.2. Generic Gradient Conflict

We now formalize the fact that the boundary gradient generically contains a normal component opposing the residual gradient.

**Assumption B.3** (Geometric Complexity of Loss Valleys in Parameter Space). We assume that the low-loss manifolds induced by the PDE residual $\mathcal{M}_{\mathrm{res}}$ form highly non-linear, curved, and geometrically complex valleys in the parameter space, as a consequence of the neural network parameterization.

**Assumption B.4** (Local Normal Preference). Assume Assumptions B.1 and B.3. For each relevant point $\bar{\theta} \in \mathcal{M}_{\mathrm{res}} \cap U$ along the *Phase II* trajectory with $\nabla_\theta \mathcal{L}_{\mathrm{res}}(\bar{\theta}) \neq 0$, define

$$v(\bar{\theta}) := \frac{\nabla_\theta \mathcal{L}_{\mathrm{res}}(\bar{\theta})}{\|\nabla_\theta \mathcal{L}_{\mathrm{res}}(\bar{\theta})\|} \in N_{\bar{\theta}} \mathcal{M}_{\mathrm{res}}. \tag{40}$$

We assume that there exist a local neighborhood $V_{\bar{\theta}} \subset U$ of $\bar{\theta}$ and a constant $\kappa_{\bar{\theta}} > 0$ such that, within $V_{\bar{\theta}}$, the boundary-loss gradient has a nonzero local preference toward one normal side of the low-residual manifold:

$$\langle \nabla_\theta \mathcal{L}_{\mathrm{bc}}(\theta), v(\bar{\theta}) \rangle \leq -\kappa_{\bar{\theta}} \quad \text{for all } \theta \in V_{\bar{\theta}}, \tag{41}$$

or

$$\langle \nabla_\theta \mathcal{L}_{\mathrm{bc}}(\theta), v(\bar{\theta}) \rangle \geq \kappa_{\bar{\theta}} \quad \text{for all } \theta \in V_{\bar{\theta}}. \tag{42}$$

In other words, near each relevant point on the *Phase II* trajectory, the boundary loss exhibits a consistent first-order tendency toward one normal side of $\mathcal{M}_{\mathrm{res}}$. We do not require this preferred side to remain fixed globally along the entire trajectory.

Assumption B.3 can be directly observed in Figures 1 and 7, as well as Table 1: the simple linear valley in function space becomes a twisted, non-convex manifold in parameter space. Assumption B.4 follows directly from continuity: whenever $\nabla_\theta \mathcal{L}_{\mathrm{bc}}(\bar{\theta})$ has a non-vanishing normal component at $\bar{\theta}$, the continuity of $\theta \mapsto \langle \nabla_\theta \mathcal{L}_{\mathrm{bc}}(\theta), v(\bar{\theta}) \rangle$ guaranties a sign-preserving neighborhood $V_{\bar{\theta}}$ with $\kappa_{\bar{\theta}} = \frac{1}{2}|\langle \nabla_\theta \mathcal{L}_{\mathrm{bc}}(\bar{\theta}), v(\bar{\theta}) \rangle|$, while the degenerate case of exact tangency is non-generic and breaks under arbitrarily small perturbations.

**Lemma B.5** (Gradient Conflict Near the Low-Residual Manifold). *Under Assumptions B.1 and B.3, consider any $\theta \in \mathcal{M}_{\mathrm{res}} \cap \mathcal{U}$ such that the boundary gradient has a non-vanishing normal component, i.e., $\nabla_\theta^\perp \mathcal{L}_{\mathrm{bc}}(\theta) \neq 0$. Then the occurrence of a negative inner product*

$$\langle \nabla_\theta \mathcal{L}_{\mathrm{res}}(\theta), \nabla_\theta \mathcal{L}_{\mathrm{bc}}(\theta) \rangle < 0 \tag{43}$$

*occurs generically along the optimization trajectory.*

*Proof.* By Lemma B.2, only the normal component of the boundary gradient $\nabla_\theta^\perp \mathcal{L}_{\mathrm{bc}}(\theta)$ contributes to the interaction. Moreover, by Assumption B.1, $\nabla_\theta \mathcal{L}_{\mathrm{res}}(\theta) \in N_\theta \mathcal{M}_{\mathrm{res}}$ is normal to $\mathcal{M}_{\mathrm{res}}$ for all $\theta \in \mathcal{M}_{\mathrm{res}} \cap U$. Let

$$v(\theta) := \frac{\nabla_\theta \mathcal{L}_{\mathrm{res}}(\theta)}{\|\nabla_\theta \mathcal{L}_{\mathrm{res}}(\theta)\|} \in N_\theta \mathcal{M}_{\mathrm{res}}, \tag{44}$$

be the unit normal given by the residual gradient at $\theta$ (when $\nabla_\theta \mathcal{L}_{\mathrm{res}}(\theta) \neq 0$; otherwise the desired strict inequality is trivial to obtain in an arbitrarily small neighborhood).

Under Assumption B.4, for each relevant point on the *Phase II* trajectory, there exists a local neighborhood in which the boundary gradient has a nonzero projection toward one normal side of $M_{\mathrm{res}}$. Hence, after possibly restricting the analysis to this local neighborhood and choosing the corresponding orientation of the normal direction, we have either $\langle \nabla_\theta L_{\mathrm{bc}}(\theta), v(\theta) \rangle < 0$ or $\langle \nabla_\theta L_{\mathrm{bc}}(\theta), v(\theta) \rangle > 0$. Without loss of generality, consider the former case (the latter is symmetric). Then

$$\langle \nabla_\theta \mathcal{L}_{\mathrm{bc}}(\theta), v(\theta) \rangle < 0. \tag{45}$$

Because $v(\theta) \in N_\theta \mathcal{M}_{\mathrm{res}}$, the inner product with $v(\theta)$ depends only on the normal projection of the boundary gradient:

$$\langle \nabla_\theta \mathcal{L}_{\mathrm{bc}}(\theta), v(\theta) \rangle = \langle \nabla_\theta^\perp \mathcal{L}_{\mathrm{bc}}(\theta), v(\theta) \rangle. \tag{46}$$

Hence,

$$\langle \nabla_\theta \mathcal{L}_{\mathrm{res}}(\theta), \nabla_\theta^\perp \mathcal{L}_{\mathrm{bc}}(\theta) \rangle = \|\nabla_\theta \mathcal{L}_{\mathrm{res}}(\theta)\| \langle v(\theta), \nabla_\theta^\perp \mathcal{L}_{\mathrm{bc}}(\theta) \rangle < 0, \tag{47}$$

which, together with Lemma B.2, completes the proof of Lemma B.5. $\square$

*Remark* B.6 (Generic Occurrence of Gradient Conflicts). In *Phase II* the iteration is confined near $\mathcal{M}_{\mathrm{res}}$ (small residual), while boundary descent attempts to move toward the boundary-preferred side (a nonzero normal component by hypothesis $\nabla_\theta^\perp \mathcal{L}_{\mathrm{bc}}(\theta) \neq 0$). As soon as the iterate is pushed to that side, $\mathcal{L}_{\mathrm{res}}$ increases, and the gradient descent update induced by the residual term, i.e., $-\nabla_\theta \mathcal{L}_{\mathrm{res}}$, drives the iterate back toward the low-residual manifold. Therefore, whenever both mechanisms are active, the residual normal direction and the boundary normal direction are opposite, producing a negative inner product.

*Remark* B.7 (Scope and Empirical Validation). In practice, owing to the complexity of neural network parameterizations, the geometry of loss valleys in the parameter space is difficult to characterize explicitly. Moreover, different PDEs and boundary conditions give rise to loss valleys with distinct geometric properties. Consequently, the above discussion focuses on high-probability behaviors in generic settings. For specific problem instances, empirical studies are required to quantitatively assess the likelihood of gradient conflicts.

## C. Aligned Constraints and Enlarged Feasible Set

In this section, we complete the theoretical justification of the aligned constraint formulation introduced in the main text. In particular, *(i)* we show that introducing an additive offset enlarges the set of admissible (low-loss) solutions; *(ii)* we interpret this effect as a thickening of the boundary-constraint set along an additive direction; and *(iii)* we discuss how this reduces the traversal required within the residual-induced loss valley.

### C.1. Enlargement of Sublevel Sets and Feasible Overlap

We now show that optimizing over $c$ can only decrease the loss, which implies that low-loss regions (sublevel sets) in parameter space become larger.

Let $u_\theta$ be a neural approximation and define (discrete) aligned losses as in the main text:

$$\mathcal{L}^{\mathrm{alg}}(\theta, c) := w_{\mathrm{res}} \mathcal{L}_{\mathrm{res}}^{\mathrm{alg}}(\theta, c) + w_{\mathrm{bc}} \mathcal{L}_{\mathrm{bc}}^{\mathrm{alg}}(\theta, c), \qquad \overline{\mathcal{L}}^{\mathrm{alg}}(\theta) := \min_{c \in \mathbb{R}} \mathcal{L}^{\mathrm{alg}}(\theta, c). \tag{48}$$

Let $\mathcal{L}^{\mathrm{std}}(\theta)$ denote the standard PINN loss obtained by fixing $c = 0$ in the same residual expressions:

$$\mathcal{L}^{\mathrm{std}}(\theta) := \mathcal{L}^{\mathrm{alg}}(\theta, 0). \tag{49}$$

**Proposition C.1** (Sublevel-Set Enlargement). *For all $\theta$, $\overline{\mathcal{L}}^{\mathrm{alg}}(\theta) \leq \mathcal{L}^{\mathrm{std}}(\theta)$. Consequently, for any $\varepsilon \geq 0$,*

$$\left\{ \theta \mid \overline{\mathcal{L}}^{\mathrm{alg}}(\theta) \leq \varepsilon \right\} \supseteq \left\{ \theta \mid \mathcal{L}^{\mathrm{std}}(\theta) \leq \varepsilon \right\}. \tag{50}$$

*Proof.* By the definition of $\overline{\mathcal{L}}^{\mathrm{alg}}(\theta)$ as a minimum over $c$,

$$\overline{\mathcal{L}}^{\mathrm{alg}}(\theta) = \min_c \mathcal{L}^{\mathrm{alg}}(\theta, c) \leq \mathcal{L}^{\mathrm{alg}}(\theta, 0) = \mathcal{L}^{\mathrm{std}}(\theta), \tag{51}$$

which proves the pointwise inequality. The sublevel-set inclusion Equation (50) follows immediately. □

### C.2. Intersection with the PDE Residual Manifold

We clarify the operational mechanism of aligned constraints and their effect on the feasible set of PDE-constrained learning.

**Assumption C.2** (Well-Posedness / Uniqueness of the Governing PDE). Assume the original PDE with the given boundary condition is well-posed and admits a unique solution $u^* \in \mathcal{F}(\Omega)$.

**Proposition C.3** (Enlarged Feasible Overlap under Aligned Constraints). *Assume that either the PDE or the boundary conditions contain at least one non-trivial zeroth-order term $\mathcal{Z}$, and that $\mathcal{Z}$ depends only on the values of $u$. Define the aligned PDE zero-residual set and boundary condition zero-residual set as*

$$\begin{aligned} \mathcal{S}_{\mathrm{res}}^{\mathrm{alg}}(c) &:= \{ u \in \mathcal{F}(\Omega) \mid \mathcal{Z}[u + c \cdot \mathbf{1}] + \mathcal{D}[u] = f \text{ in } \Omega \}, \\ \mathcal{S}_{\mathrm{bc}}^{\mathrm{alg}}(c) &:= \{ u \in \mathcal{F}(\Omega) \mid \alpha(u + c \cdot \mathbf{1}) + \beta \nabla u \cdot \mathbf{n} = g \text{ on } \Gamma \}. \end{aligned} \tag{52}$$

*Under Assumption C.2, since the original PDE has a unique solution $u^* \in \mathcal{S}_{\mathrm{res}} \cap \mathcal{S}_{\mathrm{bc}}$, then*

$$\bigcup_{c \in \mathbb{R}} \left( \mathcal{S}_{\mathrm{res}}^{\mathrm{alg}}(c) \cap \mathcal{S}_{\mathrm{bc}}^{\mathrm{alg}} \right) = \{ u^* - c \cdot \mathbf{1} \,|\, c \in \mathbb{R} \}, \tag{53}$$

*which implies that the aligned constraints enlarge the intersection of the PDE and boundary-admissible solution sets $\mathcal{S}_{\mathrm{res}}$ and $\mathcal{S}_{\mathrm{bc}}$ into an affine subspace obtained by linear shifts of the original solution, analogous to the solution structure of PDEs without zeroth-order terms under pure Neumann boundary conditions.*

*Proof.* Under Assumption C.2, if the original problem is well-posed with unique solution $u^* \in \mathcal{S}_{\mathrm{res}} \cap \mathcal{S}_{\mathrm{bc}}$, then for any $u \in \mathcal{S}_{\mathrm{res}}^{\mathrm{alg}}(c) \cap \mathcal{S}_{\mathrm{bc}}^{\mathrm{alg}}$ the reconstructed field $\tilde{u} = u + c \cdot \mathbf{1}$ satisfies the PDE and boundary condition by definition of $\mathcal{S}_{\mathrm{res}}^{\mathrm{alg}}$ and $\mathcal{S}_{\mathrm{bc}}^{\mathrm{alg}}$. Since derivative terms are invariant under constant shifts,

$$D[\tilde{u}] = D[u], \qquad \beta \nabla \tilde{u} \cdot \mathbf{n} = \beta \nabla u \cdot \mathbf{n}, \tag{54}$$

and hence

$$Z[\tilde{u}] + D[\tilde{u}] = f, \qquad \alpha \tilde{u} + \beta \nabla \tilde{u} \cdot \mathbf{n} = g, \tag{55}$$

so $\tilde{u} \in \mathcal{S}_{\mathrm{res}} \cap \mathcal{S}_{\mathrm{bc}} = \{u^*\}$ and $u = u^* - c \cdot \mathbf{1}$. $\square$

**Corollary C.4** (Aligned Minima Contain a Constant-Mode Valley in Parameters). *Under Assumption C.2, the reconstructed function $\widetilde{u}_{\theta,c}$ is unique at global optimality, but the parameter pair $(\theta, c)$ need not be. In particular, if the network class is closed under constant shifts (or approximately so), then there may exist multiple $(\theta, c)$ pairs producing the same $\widetilde{u}_{\theta,c} = u^*$, creating a connected minimizer set in $(\theta, c)$-space, and hence a wider low-loss region in $\theta$ after optimizing out $c$.*

*Remark* C.5 (Implications for Optimization). In parameter space, aligned constraints transform a low-dimensional manifold intersection problem into the search over a neighborhood with an explicit degree of freedom given by $c$. For each near-residual-minimizing parameter $\theta$, an optimal offset $c^*(\theta)$ can be chosen to reduce boundary violations, thereby improving robustness and convergence of training.

# Part II

# Methodological Supplement

## D. Optimal Offset for Zeroth-Order Terms

### D.1. Linear Case

In this section, we derive the closed-form solution for constant offset $c$ when both the PDE and the boundary conditions have zeroth-order terms that are linear operators. We work with the discrete residuals evaluated at interior points $\{x_i\}_{i=1}^{N_{\text{res}}} \subset \Omega$ and boundary points $\{x_b\}_{b=1}^{N_{\text{bc}}} \subset \Gamma$. Assume the zeroth-order PDE term is linear in $u$ at each interior collocation point:

$$\mathcal{Z}[u](x_i) = \gamma_i\, u(x_i), \qquad i = 1, \ldots, N_{\text{res}}. \tag{56}$$

Let the boundary zeroth-order coefficient be $\alpha_b$ at $x_b$ (consistent with the unified boundary form). After introducing the additive offset $c \in \mathbb{R}$ only inside zeroth-order terms, the aligned residuals become affine in $c$:

$$\bar{r}_i(c) = r_i + \gamma_i c, \qquad \bar{s}_b(c) = s_b + \alpha_b c. \tag{57}$$

The corresponding weighted aligned objective is

$$\mathcal{J}(c) := w_{\text{res}} \frac{1}{N_{\text{res}}} \sum_{i=1}^{N_{\text{res}}} |\bar{r}_i(c)|^2 + w_{\text{bc}} \frac{1}{N_{\text{bc}}} \sum_{b=1}^{N_{\text{bc}}} |\bar{s}_b(c)|^2$$

$$= w_{\text{res}} \frac{1}{N_{\text{res}}} \sum_{i=1}^{N_{\text{res}}} \left(r_i + \gamma_i c\right)^2 + w_{\text{bc}} \frac{1}{N_{\text{bc}}} \sum_{b=1}^{N_{\text{bc}}} \left(s_b + \alpha_b c\right)^2. \tag{58}$$

Differentiate with respect to $c$:

$$\mathcal{J}'(c) = 2 w_{\text{res}} \frac{1}{N_{\text{res}}} \sum_{i=1}^{N_{\text{res}}} \gamma_i \left(r_i + \gamma_i c\right) + 2 w_{\text{bc}} \frac{1}{N_{\text{bc}}} \sum_{b=1}^{N_{\text{bc}}} \alpha_b \left(s_b + \alpha_b c\right)$$

$$= 2 w_{\text{res}} \frac{1}{N_{\text{res}}} \sum_{i=1}^{N_{\text{res}}} \gamma_i r_i + 2 w_{\text{res}} \frac{1}{N_{\text{res}}} \sum_{i=1}^{N_{\text{res}}} \gamma_i^2\, c + 2 w_{\text{bc}} \frac{1}{N_{\text{bc}}} \sum_{b=1}^{N_{\text{bc}}} \alpha_b s_b + 2 w_{\text{bc}} \frac{1}{N_{\text{bc}}} \sum_{b=1}^{N_{\text{bc}}} \alpha_b^2\, c. \tag{59}$$

Setting $\mathcal{J}'(c) = 0$ gives the normal equation, we have

$$0 = w_{\text{res}} \frac{1}{N_{\text{res}}} \sum_{i=1}^{N_{\text{res}}} \gamma_i r_i + w_{\text{res}} \frac{1}{N_{\text{res}}} \sum_{i=1}^{N_{\text{res}}} \gamma_i^2\, c + w_{\text{bc}} \frac{1}{N_{\text{bc}}} \sum_{b=1}^{N_{\text{bc}}} \alpha_b s_b + w_{\text{bc}} \frac{1}{N_{\text{bc}}} \sum_{b=1}^{N_{\text{bc}}} \alpha_b^2\, c. \tag{60}$$

Collecting the terms that multiply $c$ and defining

$$A := w_{\text{res}} \frac{1}{N_{\text{res}}} \sum_{i=1}^{N_{\text{res}}} \gamma_i^2 + w_{\text{bc}} \frac{1}{N_{\text{bc}}} \sum_{b=1}^{N_{\text{bc}}} \alpha_b^2, \tag{61}$$

then we can rewrite Equation (60) as

$$A\, c = -w_{\text{res}} \frac{1}{N_{\text{res}}} \sum_{i=1}^{N_{\text{res}}} \gamma_i r_i - w_{\text{bc}} \frac{1}{N_{\text{bc}}} \sum_{b=1}^{N_{\text{bc}}} \alpha_b s_b. \tag{62}$$

When $A > 0$, the quadratic $\mathcal{J}(c)$ is strictly convex because

$$\mathcal{J}''(c) = 2A > 0, \tag{63}$$

and therefore the stationary point is the unique global minimizer. Solving Equation (62) yields the closed-form optimal offset

$$c^* = -\frac{w_{\text{res}} \frac{1}{N_{\text{res}}} \sum_{i=1}^{N_{\text{res}}} \gamma_i r_i + w_{\text{bc}} \frac{1}{N_{\text{bc}}} \sum_{b=1}^{N_{\text{bc}}} \alpha_b s_b}{w_{\text{res}} \frac{1}{N_{\text{res}}} \sum_{i=1}^{N_{\text{res}}} \gamma_i^2 + w_{\text{bc}} \frac{1}{N_{\text{bc}}} \sum_{b=1}^{N_{\text{bc}}} \alpha_b^2}. \tag{64}$$

## D.2. Nonlinear Case

In this section, we consider the case where the zeroth-order PDE operator $\mathcal{Z}[\cdot]$ is nonlinear in $u$. As before, the network output $u_\theta$ is treated as fixed when optimizing the constant offset $c$. After introducing $c$ only inside zeroth-order terms, the aligned residuals $\bar{r}_i(c)$ and $\bar{s}_b(c)$ are generally nonlinear functions of $c$. The weighted aligned objective $\mathcal{J}(c)$ is defined as in Equation (58), but no closed-form minimizer exists in this case. We therefore apply Newton's method for several steps to update $c$:

$$c_{k+1} = c_k - \frac{\mathcal{J}'(c_k)}{\mathcal{J}''(c_k)}, \tag{65}$$

where $\mathcal{J}'(c_k)$ and $\mathcal{J}''(c_k)$ is calculated by the automatic differentiation tool of PyTorch.

In practice, we use a small number of Newton inner iterations. To stabilize early training, we perform more inner steps $K_{\text{init}}$ in the first optimization iteration, and then reduce to one or two steps $K_{\text{few}}$ afterwards. After the training has stabilized and converged ($t \geq t_c$), we fix the value of $c$ and no longer update it in order to further improve accuracy. Throughout, $c$ is treated as a constant during backpropagation, and gradients are computed only with respect to $u_\theta$.

We briefly analyze the local non-degeneracy of the 1D subproblem $\mathcal{J}(c)$ defined in Equation (14), which underlies the local convergence of Newton's iteration in Equation (65). Differentiating $\mathcal{J}$ twice with respect to $c$ gives

$$\mathcal{J}''(c) = \frac{2w_{\text{res}}}{N_{\text{res}}} A(c) + \frac{2w_{\text{bc}}}{N_{\text{bc}}} B(c), \tag{66}$$

where $A(c) := \sum_i [\bar{r}_i'(c)^2 + \bar{r}_i(c)\bar{r}_i''(c)]$ and $B(c) := \sum_b [\bar{s}_b'(c)^2 + \bar{s}_b(c)\bar{s}_b''(c)]$, with primes denoting derivatives with respect to $c$.

For Dirichlet, Neumann, and Robin boundaries, $\bar{s}_b(c)$ is affine in $c$, so $B(c) = \sum_b \bar{s}_b'(c)^2 \geq 0$. The only potential source of negative curvature is the cross term $\sum_i \bar{r}_i \bar{r}_i''$ in $A(c)$, which we cannot guarantee to be non-negative for arbitrary nonlinear $\mathcal{Z}$. In practice, however, the signed quantities $\bar{r}_i \bar{r}_i''$ exhibit partial cancellation across collocation points, whereas $\bar{r}_i'^2$ accumulates with $N_{\text{res}}$, so $\mathcal{J}''(c^\star) > 0$ holds in the regimes encountered during training and Newton's method retains its quadratic local convergence rate. In the rare case where this fails, the Newton step can be safeguarded by a line search, or replaced by first-order alternatives such as gradient descent or the secant method.

# E. Pipeline

---
**Algorithm 1** CAML: Constraint-Aligned Loss with Manifold Lifting
---

**Input:** PDE operator $\mathcal{N}$, boundary operator $\mathcal{B}$, network $u_\theta$
**Parameters:** weights $w_{\text{res}}, w_{\text{bc}}$, delay schedule $\lambda(t)$
Initialize network parameters $\theta$
**for** training step $t = 1$ **to** $T$ **do**
    Sample interior points $\{x_i\}$ and boundary points $\{x_b\}$
    Compute residuals:
$$r_i = \mathcal{N}[u_\theta](x_i) - f(x_i), \quad s_b = \mathcal{B}[u_\theta](x_b) - g(x_b)$$
    Store all derivative terms $\mathcal{D}[u_\theta](x_i)$ and $\beta_b \nabla u_\theta(x_b) \cdot \mathbf{n}_b$ temporarily
    **Solve offset $c$:**
    **if** zeroth-order terms are linear **then**
        Compute $c$ by closed-form weighted least squares
    **else**
        $K \leftarrow K_{\text{few}} \cdot \mathbb{I}(t < t_c) + K_{\text{init}} \cdot \mathbb{I}(t = 1)$
        Update $c$ using $K$ Newton steps on $\mathcal{L}(c)$
    **end if**
    Apply aligned residuals $\bar{r}_i = r_i(u_\theta + c), \bar{s}_b = s_b(u_\theta + c)$, where all derivative terms are directly loaded from cache
    $\mathcal{L} = w_{\text{res}}\lambda(t)\mathcal{L}_{\text{res}}^{\text{alg}} + w_{bc}\mathcal{L}_{\text{bc}}^{\text{alg}}$
    Update $\theta \leftarrow \theta - \eta\nabla_\theta\mathcal{L}$
**end for**

---

## F. Discussion and Future Work

We choose to align the model and solution at constant-mode because it provides the best benefit-to-cost trade-off for the common types of PDE, as it relaxes both the PDE residual and the boundary constraints while keeping the inner solve 1D (closed-form in linear cases). Furthermore, it ingeniously utilizes the additive invariance of the derivative terms, thus allowing the calculation of $c$ to only consider the influence of zeroth-order terms. Unless for special PDEs, extending the lifting to nontrivial modes would require incorporating derivative terms into the Newton updates and computing higher-order/mixed auto differentiation throughout the network at every step, which is prohibitively expensive and numerically fragile in practice. Therefore, it is difficult to quantify its benefits across different physical fields.

Nevertheless, we do not rule out the possibility that, for certain special classes of PDEs, there exist alignment modes that are both inexpensive and potentially more beneficial. Future work may systematically characterize the structural features of these low-cost, high-gain modes and explore how constant-mode alignment can be extended to broader yet still low-dimensionally parameterized alignment spaces without compromising numerical stability.

# Part III

# Experimental Supplement

## G. Loss Valley Visualization

In this section, we detail how we generate the two subfigures in Figure 1, which visualize the PDE-residual loss landscape projected onto a 2D subspace, in *(i)* the function space and *(ii)* the parameter space. The goal is to highlight that, although the residual-induced valley is relatively simple in the function space, it becomes highly distorted and non-convex after being mapped through a neural parameterization, consistent with the discussion in Section 4.1.1.

We consider a one-dimensional Poisson-type equation on the interval $[0, \pi]$:

$$u''(x) = -\sin(x), \quad x \in [0, \pi]. \tag{67}$$

Given a candidate solution $u(x)$, the residual loss over a set of $N$ collocation points $\{x_i\}_{i=1}^N$ is

$$\mathcal{L}_{\text{res}}(u) = \frac{1}{N} \sum_{i=1}^{N} |u''(x) + \sin(x)|^2. \tag{68}$$

For all experiments in Figure 1, we use $N = 100$ uniformly spaced collocation points on $[0, \pi]$.

### G.1. Figure 1(a): Loss Landscape in Function Space

To visualize the loss landscape in the function space, each candidate solution $u(x)$ is represented by its values at the collocation points:

$$\mathbf{u} = [u(x_1), u(x_2), \ldots, u(x_N)]^\top \in \mathbb{R}^N. \tag{69}$$

Thus, the infinite-dimensional function space is approximated by an $N$-dimensional Euclidean space with $N = 100$.

We generate candidate solutions using a simple sinusoidal ansatz

$$u_\theta(x) = A\sin(\omega x + \phi) + B, \tag{70}$$

where $\theta = (\omega, \phi, A, B)$.

Three solutions are constructed by directly setting the parameters:

$$(\omega, \phi, A, B) \in \{(1, 0, 1, 0), (1, 0, 1, -1), (1, 0, 1, 1)\}. \tag{71}$$

Each solution is evaluated on the collocation grid, yielding three vectors $\mathbf{u}_1, \mathbf{u}_2, \mathbf{u}_3 \in \mathbb{R}^{100}$. This parameter combination ensures that different solutions only differ in terms of the additive constant $B$. In the function space, it forms a straight zero-residual manifold along the dimension of $B$, which is the loss valley.

We define a two-dimensional affine subspace in $\mathbb{R}^{100}$ spanned by the three solution vectors. The center of the plane is chosen as $\mathbf{c} = \mathbf{u}_1$. The two orthonormal directions are constructed via Gram–Schmidt:

$$\mathbf{d}_1 = \frac{\mathbf{u}_2 - \mathbf{u}_1}{\|\mathbf{u}_2 - \mathbf{u}_1\|}, \qquad \mathbf{d}_2 = \frac{(\mathbf{u}_3 - \mathbf{u}_1) - \langle \mathbf{u}_3 - \mathbf{u}_1, \mathbf{d}_1 \rangle \mathbf{d}_1}{\|(\mathbf{u}_3 - \mathbf{u}_1) - \langle \mathbf{u}_3 - \mathbf{u}_1, \mathbf{d}_1 \rangle \mathbf{d}_1\|}. \tag{72}$$

Any point on this plane is parameterized by scalars $(\alpha, \beta)$:

$$\mathbf{u}(\alpha, \beta) = \mathbf{c} + \alpha \mathbf{d}_1 + \beta \mathbf{d}_2. \tag{73}$$

Since only the discretized values $\mathbf{u}(\alpha, \beta)$ are available, the second derivative $u''(x)$ is approximated using second-order finite differences on the uniform grid. The resulting landscape is visualized using contour plots with logarithmic scaling to capture variations across multiple orders of magnitude. The red curve in Figure 1(a) illustrates a low-loss path within this projected landscape.

## G.2. Figure 1(b): Loss Landscape in Parameter Space

To visualize Figure 1(b), we employ a fully connected feedforward neural network with three hidden layers. The network is trained under three analytical solutions defined by Section G.1, and each instance is optimized until convergence to a very low residual loss. Specifically, the loss function we use is composed of the following elements:

$$\mathcal{L}(\theta) = \mathcal{L}_{\text{res}} + \mathcal{L}_{\text{data}}, \tag{74}$$

where $\mathcal{L}_{\text{data}}$ is the solution value obtained from Equation (70).

The experimental hyperparameters are shown in Table 6.

*Table 6.* Hyperparameters used for training the neural networks.

| Category | Hyperparameter | Value |
|---|---|---|
| Network Architecture | Number of hidden layers | 3 |
| | Neurons per hidden layer | 20 |
| | Activation function | `tanh` |
| Training Setup | Optimizer | Adam |
| | Learning rate | $1 \times 10^{-3}$ |
| | Batch size | Full-batch |
| | Total training steps | 3000 |

We denote the resulting parameter configurations by $\theta_1, \theta_2, \theta_3 \in \mathbb{R}^P$, corresponding to the three independently trained models. A two-dimensional affine subspace in parameter space is constructed with center $\theta_c = \theta_1$ and directions

$$\mathbf{v}_1 = \frac{\theta_2 - \theta_1}{\|\theta_2 - \theta_1\|}, \qquad \mathbf{v}_2 = \frac{(\theta_3 - \theta_1) - \langle\theta_3 - \theta_1, \mathbf{v}_1\rangle\mathbf{v}_1}{\|(\theta_3 - \theta_1) - \langle\theta_3 - \theta_1, \mathbf{v}_1\rangle\mathbf{v}_1\|}. \tag{75}$$

Any parameter point on this plane is given by

$$\theta(\alpha, \beta) = \theta_c + \alpha\mathbf{v}_1 + \beta\mathbf{v}_2. \tag{76}$$

Although the residual loss valley is relatively simple and flat in the function space, the nonlinear mapping from parameters $\theta$ to functions $u_\theta(\cdot)$ can significantly distort this structure. Consequently, the projected loss landscape in parameter space exhibits curvature, folding, and local non-convexity, as illustrated in Figure 1(b).

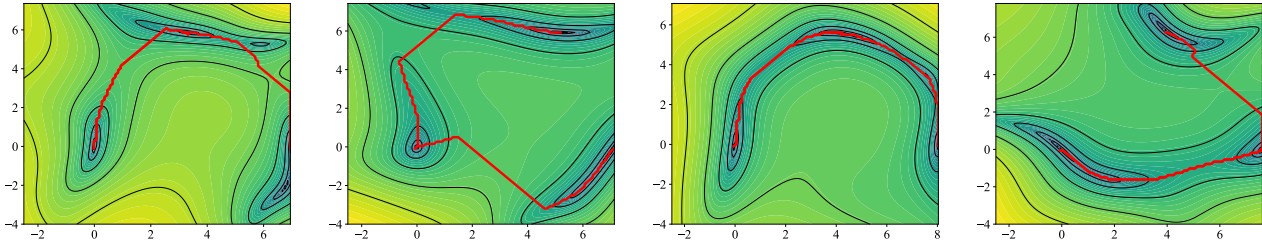

*Figure 7.* Visualization of the loss valleys in the parameter space under 4 group of different random seeds.

We also conducted experiments with more random initializations, as shown in Figure 7. Note that due to the limitation of two-dimensional slicing, Figure 7 provides a cross-sectional view of the high-dimensional loss landscape, only revealing the rough trajectory of the loss valley rather than indicating that all points along the valley bottom attain zero loss.

## G.3. Table 1: Data Statistics

We record and analyze the Hessian from the above two experiments in both function and parameter space in Table 1. Let $f : \mathbb{R}^d \to \mathbb{R}$ be twice continuously differentiable. The condition number of the Hessian matrix $H(x) = \nabla^2 f(\theta)$ (at a point

$\theta$) is defined as

$$\kappa(H(\theta)) \triangleq \|H(\theta)\|_2 \|H(\theta)^{-1}\|_2 = \frac{\lambda_{\max}(H(\theta))}{\lambda_{\min}(H(\theta))}, \tag{77}$$

where $\lambda_{\max}(\cdot)$ and $\lambda_{\min}(\cdot)$ denote the largest and smallest eigenvalues, respectively. A larger condition number indicates a more ill-conditioned loss landscape, characterized by elongated, valley-like level sets. In contrast, when the condition number is small, the contour lines become nearly spherical, suggesting that the curvature is more uniform across different directions.

We also compute the subspace similarity of the Hessian to quantify the consistency of low-curvature directions across different constants. Let $V_i \in \mathbb{R}^{d \times k}$ denote the matrix whose columns are the $k$ eigenvectors corresponding to the smallest-magnitude eigenvalues of the Hessian at solution $i$, forming a near-zero (low-curvature) eigenspace. For the function space, we set $k = 1$, and for the parameter space, we set $k = 100$. The subspace similarity between solutions $i$ and $j$ is defined as

$$\mathrm{Sim}(V_i, V_j) \triangleq \frac{\|V_i^\top V_j\|_F}{\sqrt{k}}, \tag{78}$$

where $\|\cdot\|_F$ denotes the Frobenius norm. This quantity lies in $[0, 1]$, with larger values indicating stronger alignment between the corresponding low-curvature subspaces. By combining the condition number and the subspace similarity, we characterize not only the degree of degeneracy of the loss landscape but also the geometric stability of its flat directions across independently trained solutions.

In the function space, the Hessian exhibits an infinite condition number, and the zero-curvature subspace is strictly aligned, indicating that the loss valley corresponds to a standard straight zero-residual manifold. In the parameter space, although exact zero residuals cannot be achieved due to training error, the condition number remains extremely large. Moreover, the similarity of the low-curvature subspaces in the parameter space stays only around 50% even when $k$ approaches one-fourth of the total number of network parameters, suggesting that the loss valley is mapped into the parameter space in a highly distorted and non-convex manner.

## H. Demonstration of Two-Phase Optimization Dynamics

In this section, we present a minimal yet reproducible numerical experiment that illustrates the Two-Phase behavior of standard PINNs discussed in Section 4.2. In particular, we demonstrate how large-amplitude Dirichlet boundary conditions can induce optimization stagnation or oscillations after the PDE residual has already been minimized.

### H.1. Experiment Setup

We consider the Poisson equation on the unit square domain $\Omega = (0, 1)^2, \Gamma = \partial\Omega$, given by

$$\Delta u(x, y) = f(x, y), \quad (x, y) \in \Omega, \tag{79}$$

with Dirichlet boundary conditions

$$u(x, y) = g(x, y), \quad (x, y) \in \Gamma. \tag{80}$$

To enable quantitative evaluation, the exact solution is constructed as

$$u^*(x, y) = g(x, y) + \sin(\pi x)\sin(\pi y), \tag{81}$$

where the boundary function $g$ is defined as

$$g(x, y) = A\sin(2\pi x) + B\cos(3\pi y) + Cx + Dy, \tag{82}$$

with coefficients $A = 20, B = 15, C = 8, D = 6$. This choice intentionally produces a boundary condition with a large amplitude range, significantly increasing the discrepancy between the randomly initialized network output and the true boundary values.

The right-hand side $f(x, y)$ is computed analytically from $u^*$ as

$$\begin{aligned} f(x, y) &= \Delta g(x, y) + \Delta\big(\sin(\pi x)\sin(\pi y)\big) \\ &= -(2\pi)^2 A\sin(2\pi x) - (3\pi)^2 B\cos(3\pi y) - 2\pi^2 \sin(\pi x)\sin(\pi y). \end{aligned} \tag{83}$$

We employ a fully connected feedforward neural network to fit this problem. The hyperparameters are shown in Table 7.

*Table 7.* Hyperparameters used for training the neural networks.

| Category | Hyperparameter | Value |
|---|---|---|
| Network Architecture | Number of hidden layers | 5 |
| | Neurons per hidden layer | 80 |
| | Activation function | `tanh` |
| Training Setup | Optimizer | Adam |
| | Learning rate | $1 \times 10^{-3}$ |
| | Batch size | Full-batch |
| | Total training steps | 6000 |
| Sample Mode | Inner collocation points | 8000 |
| | Boundary collocation points | $600 \times 4$ |

## H.2. Observed Two-Phase Behavior

To characterize this failure mode more quantitatively, we recommend monitoring the following diagnostics: *(a)* The evolution of $\mathcal{L}_{\mathrm{res}}$, *(b)* $\mathcal{L}_{\mathrm{bc}}$, *(c)* the cosine similarity between task gradients $\cos(\phi)$, and *(d)* gradient norm ratio $\rho_g = \frac{\|\nabla_\theta \mathcal{L}_{\mathrm{res}}\|}{\|\nabla_\theta \mathcal{L}_{\mathrm{bc}}\|}$. Under the above setting, standard PINNs frequently exhibit two-phase behavior, as shown in Figure 8. We have also visualized the optimization trajectory during the training process and superimposed it with the loss landscape, enabling a more intuitive observation of the optimization dynamics, as shown in Figure 9.

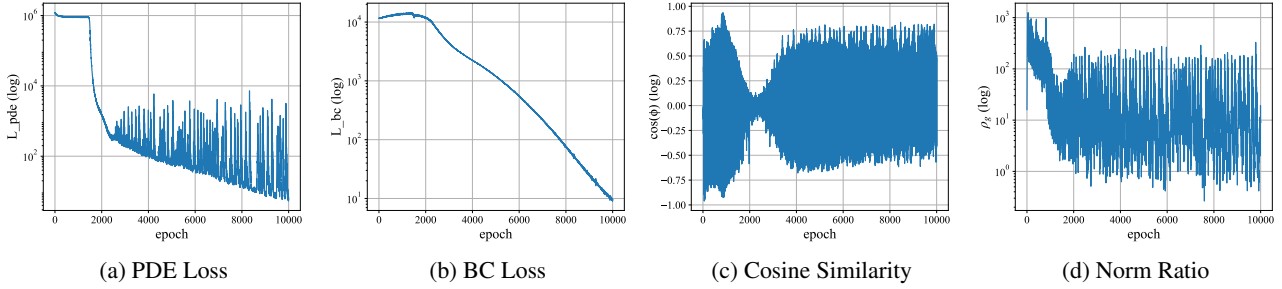

|   (a) PDE Loss   |   (b) BC Loss   |   (c) Cosine Similarity   |   (d) Norm Ratio   |

*Figure 8.* Evolution of loss components and gradient statistics during PINN training, including the PDE residual loss, boundary condition loss, cosine similarity between their gradients, and the parameter optimize trajectory.

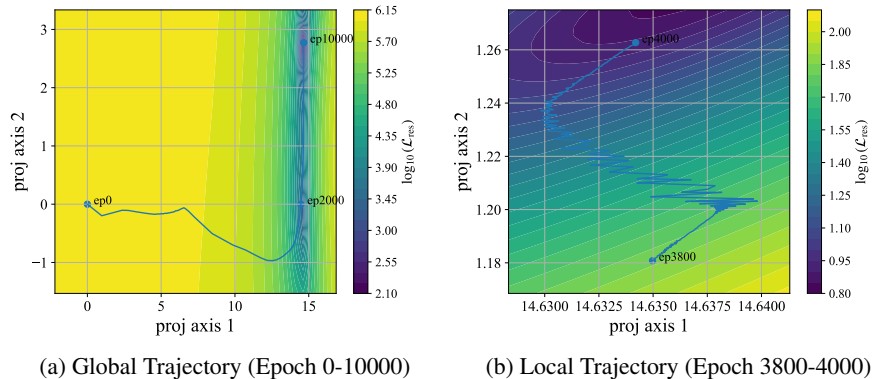

|   (a) Global Trajectory (Epoch 0-10000)   |   (b) Local Trajectory (Epoch 3800-4000)   |

*Figure 9.* Visualization of parameter trajectories and the corresponding PDE residual loss landscape, from both (a) global and (b) local perspective. While the global loss curve appears nearly linear in loss valley, the local optimization path displays substantial oscillatory behavior, indicating persistent gradient conflicts within the residual-induced valley.

**Phase I: Rapid PDE Residual Minimization (0–2000 epochs).** *(a) PDE residual loss*: Decreases rapidly by several orders

of magnitude and quickly reaches a near-minimal level, indicating that the optimization is strongly attracted to the loss valley induced by the zero-residual PDE manifold. *(b) Boundary condition loss*: Exhibits transient growth or noticeable fluctuations, suggesting that boundary constraints are not yet effectively enforced during this phase. *(c) Gradient angle cosine similarity*: Displays large fluctuations whose amplitude gradually diminishes toward zero, reflecting a weakening interaction between the two gradient components. *(d) Gradient norm ratio*: Significantly greater than one and decreases steadily, confirming the dominant influence of the PDE residual gradient in early training.

**Phase II: Boundary-Dominated Valley Traversal (2000–10000 epochs).** *(a) PDE residual loss*: Ceases to decrease monotonically and instead exhibits pronounced oscillations, characteristic of optimization dynamics within a high-curvature region near the bottom of the residual-induced valley. *(b) Boundary condition loss*: Continues to decrease smoothly and steadily, indicating that boundary constraints increasingly govern the optimization trajectory. *(c) Gradient angle cosine similarity*: Shows a brief interval of near-zero values immediately after the PDE residual reaches its minimum, implying near-orthogonality between PDE and boundary gradients. Subsequently, boundary gradients begin to steer the optimization; however, the rugged geometry of the residual valley reintroduces significant oscillations. *(d) Gradient norm ratio*: The mean value is significantly lower than in *Phase I*, while oscillation amplitude increases substantially, signaling that boundary conditions and PDE residual frequently compete for control of the gradient direction.

# I. Main Experimental Setup

## I.1. Benchmarks

**Heat.** We consider a 2D steady-state heat conduction benchmark on a square domain $\Omega = [0,1] \times [0,1]$. The goal is to learn/solve a scalar temperature field $u(x,y)$ that satisfies a homogeneous elliptic PDE in the interior and mixed (Dirichlet/Neumann) boundary conditions on $\Gamma$. This benchmark evaluates the ability of a model to handle mixed boundary conditions.

Inside the domain, $u$ satisfies the Laplace equation

$$\Delta u(x,y) \; = \; \frac{\partial^2 u}{\partial x^2} + \frac{\partial^2 u}{\partial y^2} \; = \; 0, \qquad (x,y) \in \Omega. \tag{84}$$

We impose Dirichlet conditions on the left and bottom boundaries,

$$u(0,y) = T_0, \quad u(x,0) = T_0, \qquad T_0 = 100, \tag{85}$$

and Neumann conditions on the right and top boundaries (outward normals $(1,0)$ and $(0,1)$),

$$\frac{\partial u}{\partial x}(L,y) = -q, \quad \frac{\partial u}{\partial y}(x,L) = -q, \qquad q = 15. \tag{86}$$

These correspond to a constant prescribed boundary temperature on $\{x=0\} \cup \{y=0\}$ and a constant outward heat flux on $\{x=L\} \cup \{y=L\}$.

The analytical solution is computed via a truncated Fourier-series representation that satisfies the above mixed boundary conditions:

$$u(x,y) = T_0 + \sum_{m=1}^{M} A_n \Big( \sinh(\lambda x)\sin(\lambda y) + \sinh(\lambda y)\sin(\lambda x) \Big), \quad n = 2m-1,\, \lambda = \frac{n\pi}{2L}, \tag{87}$$

with coefficients

$$A_n = \frac{-8qL}{k\, n^2 \pi^2 \cosh\left(\frac{n\pi}{2}\right)}, \qquad (k=1), \tag{88}$$

and we use $M = 20$ terms in practice to obtain an accurate numerical approximation of the analytical series solution.

**Poisson.** We consider a 2D Poisson benchmark on the unit square $\Omega = [0,1] \times [0,1]$. The task is to solve for a scalar field $u(x,y)$ governed by a Poisson equation with a prescribed source term and Dirichlet boundary conditions on the entire boundary $\Gamma$. This benchmark evaluates the ability of a model to handle complex and strong nonlinear boundary conditions.

Inside the domain, $u$ satisfies

$$\Delta u(x, y) = f(x, y), \qquad (x, y) \in \Omega, \tag{89}$$

where the source term is defined as

$$f(x, y) = -(2\pi)^2 A \sin(2\pi x) - (3\pi)^2 B \cos(3\pi y) - 2\pi^2 \sin(\pi x) \sin(\pi y), \tag{90}$$

with constants $A = 30$, $B = 25$.

Dirichlet boundary conditions are imposed on the whole boundary:

$$u(x, y) = g(x, y), \qquad (x, y) \in \Gamma, \tag{91}$$

where

$$g(x, y) = A \sin(2\pi x) + B \cos(3\pi y) + Cx + Dy + 10, \tag{92}$$

and $C = 18$, $D = 16$.

This Poisson problem is constructed via the method of manufactured solutions. The exact solution is given by

$$u^*(x, y) = g(x, y) + \sin(\pi x) \sin(\pi y), \tag{93}$$

which satisfies the prescribed Dirichlet boundary conditions and yields the source term $f(x, y) = \Delta u^*(x, y)$. This analytical solution is used as the ground truth for quantitative error evaluation.

**Navier–Stokes (NS).** We consider a 2D steady incompressible Navier–Stokes benchmark on the unit square $\Omega = [0, 1] \times [0, 1]$. The objective is to solve for the velocity field $\boldsymbol{u}(x, y) = (u(x, y), v(x, y))$ and the pressure field $p(x, y)$ governed by the steady Navier–Stokes equations with a prescribed forcing term and Dirichlet boundary conditions. This benchmark evaluates the ability of a model to handle nonlinear convection, pressure–velocity coupling, and incompressibility constraints.

Inside the domain, the steady incompressible Navier–Stokes equations are given by

$$\begin{aligned}
(\boldsymbol{u} \cdot \nabla)\boldsymbol{u} + \nabla p - \frac{1}{\text{Re}} \Delta \boldsymbol{u} &= \boldsymbol{f}(x, y), & (x, y) \in \Omega, \\
\nabla \cdot \boldsymbol{u} &= 0, & (x, y) \in \Omega,
\end{aligned} \tag{94}$$

where $\text{Re} = 500$ denotes the Reynolds number and $\boldsymbol{f} = (f_u, f_v)$ is a known forcing term constructed so that the system admits a closed-form analytical solution.

Dirichlet boundary conditions are imposed on the entire boundary:

$$\boldsymbol{u}(x, y) = \boldsymbol{u}^*(x, y), \qquad (x, y) \in \Gamma, \tag{95}$$

where $\boldsymbol{u}^*$ is the prescribed exact velocity field. No explicit boundary condition is imposed on the pressure.

The benchmark is defined using a manufactured solution. Let

$$\begin{aligned}
u^*(x, y) &= \pi \sin(\pi x) \cos(\pi y) + U_x, \\
v^*(x, y) &= -\pi \cos(\pi x) \sin(\pi y) + U_y, \\
p^*(x, y) &= \sin(2\pi x) \sin(2\pi y),
\end{aligned} \tag{96}$$

where $(U_x = 1, U_y = 1)$ is a constant velocity offset. The forcing term $\boldsymbol{f}$ is analytically derived by substituting $(\boldsymbol{u}^*, p^*)$ into the Navier–Stokes equations. This exact solution satisfies the incompressibility condition and the prescribed Dirichlet boundary conditions, and is used as the ground truth for error evaluation.

**Helmholtz.** We consider a 2D Helmholtz-type benchmark defined on a bounded, irregular domain $\Omega \subset \mathbb{R}^2$. The task is to solve for a scalar field $u(x, y)$ governed by a variable-coefficient second-order elliptic equation with a reaction term. This benchmark tests models' ability to handle spatially varying anisotropic operators and nontrivial domain geometries.

Inside the domain, $u$ satisfies a generalized Helmholtz equation of the form

$$-\nabla \cdot \big(A(x,y)\nabla u(x,y)\big) + q(x,y)\, u(x,y) = f(x,y), \qquad (x,y) \in \Omega, \tag{97}$$

where the diffusion tensor $A(x, y) \in \mathbb{R}^{2 \times 2}$ and the reaction coefficient $q(x, y)$ are given by

$$A(x,y) = \begin{pmatrix} 1 + 0.3x & 0.15\sin(\pi x)\sin(\pi y) \\ 0.15\sin(\pi x)\sin(\pi y) & 1 + 0.3y \end{pmatrix},$$
$$q(x,y) = 2 + \cos(\pi x)\cos(\pi y). \tag{98}$$

Dirichlet boundary conditions are imposed on the entire boundary:

$$u(x,y) = u^*(x,y), \qquad (x,y) \in \Gamma, \tag{99}$$

where $u^*$ is a prescribed exact solution.

The benchmark is constructed using a manufactured solution,

$$u^*(x,y) = U_0 + \sin(\pi x)\cos(2\pi y) + 0.2\, e^{x+y}, \qquad U_0 = 100. \tag{100}$$

The source term $f(x, y)$ is obtained analytically by substituting $u^*$ into the governing equation. This exact solution satisfies the Dirichlet boundary conditions and serves as the ground truth for quantitative error evaluation.

## I.2. Model and Training Configuration

Among all the benchmarks and baselines, the configurations of MLP, PirateNets, and PINNsFormer we used are shown in Table 8, Table 9, and Table 10, respectively.

*Table 8.* Configuration of MLP across all benchmarks.

| Configuration | Value |
|---|---|
| Number of hidden layers | 4 |
| Neurons per hidden layer | 64 |
| Activation function | `tanh` |

*Table 9.* Configuration of PirateNets across all benchmarks.

| Configuration | Value |
|---|---|
| Number of hidden layers | 3 |
| Neurons per hidden layer | 32 |
| Activation function | `tanh` |

*Table 10.* Configuration of PINNsFormer across all benchmarks.

| Configuration | Value |
|---|---|
| Number of attention layers in encoder | 2 |
| Number of attention layers in decoder | 2 |
| Head number in attention layer | 1 |
| Neurons per hidden layer | 32 |
| Activation function | `WaveAct` |

For PINNsFormer in all benchmarks and baselines, we detach the gradients at the encoder input to prevent gradient mixing among neighboring spatial points when automatic differentiation computes the vector–Jacobian product.

Among all the benchmarks and baselines, the hyperparameters we used to train MLP, PirateNets, and PINNsFormer are shown in Table 11, Table 12, and Table 13, respectively.

*Table 11.* Hyperparameters of MLP across all benchmarks. Where $\eta$ is the learning rate, $w_{res}$ is the PDE residual loss weight, $w_{bc}$ is the boundary condition loss weight, $T_{min}$ is the minimum number of iterations, $T_{max}$ is the maximum number of iterations, and $L_2^{stop}$ is the precision threshold for terminating training.

| Benchmark | $\eta$ | $w_{res}$ | $w_{bc}$ | $T_{min}$ | $T_{max}$ | $L_2^{stop}$ |
|---|---|---|---|---|---|---|
| Heat | 1.0e-3 | 1 | 5 | 6000 | 20000 | 2.0e-3 |
| Poisson | 1.0e-3 | 1 | 100 | 6000 | 20000 | 1.0e-2 |
| NS | 1.0e-3 | 1 | 100 | 6000 | 20000 | 5.0e-3 |
| Helmholtz | 1.0e-3 | 1 | 10 | 4000 | 20000 | 1.0e-3 |

*Table 12.* Hyperparameters of PirateNets across all benchmarks. Where $\eta$ is the learning rate, $w_{res}$ is the PDE residual loss weight, $w_{bc}$ is the boundary condition loss weight, $T_{min}$ is the minimum number of iterations, $T_{max}$ is the maximum number of iterations, and $L_2^{stop}$ is the precision threshold for terminating training.

| Benchmark | $\eta$ | $w_{res}$ | $w_{bc}$ | $T_{min}$ | $T_{max}$ | $L_2^{stop}$ |
|---|---|---|---|---|---|---|
| Heat | 1.0e-3 | 1 | 1 | 6000 | 20000 | 1.0e-2 |
| Poisson | 1.0e-3 | 1 | 100 | 6000 | 20000 | 5.0e-2 |
| NS | 3.0e-3 | 1 | 100 | 6000 | 20000 | 1.0e-2 |
| Helmholtz | 1.0e-3 | 1 | 100 | 4000 | 20000 | 5.0e-3 |

*Table 13.* Hyperparameters of PINNsFormer across all benchmarks. Where $\eta$ is the learning rate, $w_{res}$ is the PDE residual loss weight, $w_{bc}$ is the boundary condition loss weight, $T_{min}$ is the minimum number of iterations, $T_{max}$ is the maximum number of iterations, and $L_2^{stop}$ is the precision threshold for terminating training.

| Benchmark | $\eta$ | $w_{res}$ | $w_{bc}$ | $T_{min}$ | $T_{max}$ | $L_2^{stop}$ |
|---|---|---|---|---|---|---|
| Heat | 1.0e-3 | 1 | 1 | 2000 | 10000 | 1.0e-2 |
| Poisson | 1.0e-3 | 1 | 100 | 2000 | 10000 | 5.0e-2 |
| NS | 1.0e-3 | 1 | 100 | 2000 | 10000 | 1.0e-2 |
| Helmholtz | 1.0e-3 | 1 | 100 | 1500 | 10000 | 1.0e-3 |

For CAML, the additional hyperparameters we adopted in different benchmarks are as in Table 14. For other baselines that have additional hyperparameters, we apply the recommended configurations as stated in their original papers.

*Table 14.* Additional hyperparameters of CAML across all benchmarks. Where $t_d$ and $t_r$ are the delay step number and the ramp length of the delay-residual gating function, $K_{init}$ and $K_{few}$ are the initial Newton steps and within-loop Newton steps of $c$ in non-linear PDE benchmark, and $t_c$ is the maximum iteration to stop update $c$ in non-linear PDE benchmark.

| Benchmark | $t_d$ | $t_r$ | $K_{init}$ | $K_{few}$ | $t_c$ |
|---|---|---|---|---|---|
| Heat | 25 | 50 | — | — | — |
| Poisson | 200 | 800 | — | — | — |
| NS | 25 | 50 | 10 | 2 | 1000 |
| Helmholtz | 25 | 50 | — | — | — |

## J. Computational Overhead

In this section, we present a quantitative analysis of computational costs for both linear and nonlinear CAML PDEs. We use the Heat benchmark as the linear case, and use NS as the nonlinear benchmark. All configurations are consistent with the main experiment. We first measure the time required to compute the closed-form solution for $c$ in the linear case, as well as the proportion of that time spent in the single iteration. The results are shown in Table 15.

*Table 15.* The computational cost of CAML in linear PDEs. FP: the forward propagation process; AD: automatic differentiation of PDE residual and boundary conditions; AC: calculation of the additive constant $c$; BP: the backward propagation process; Percentage: The proportion of time spent on calculating $c$ in the total duration.

|  |  | FP | AD | **AC** | BP | Percentage |
|---|---|---|---|---|---|---|
| MLP | Single Iteration ($ms$) | 0.1619 | 1.6322 | **0.1439** | 2.6541 | 3.13% |
|  | Full Process ($s$) | 1.1971 | 10.7513 | **0.9939** | 17.2778 | 3.29% |
| PirateNets | Single Iteration ($ms$) | 0.7503 | 7.0681 | **0.1763** | 15.4657 | 0.75% |
|  | Full Process ($s$) | 6.1116 | 57.5850 | **1.1378** | 116.7246 | 0.63% |
| PINNsFormer | Single Iteration ($ms$) | 2.3134 | 21.4542 | **0.1487** | 47.0843 | 0.21% |
|  | Full Process ($s$) | 6.8850 | 66.9518 | **0.4497** | 140.9999 | 0.21% |

It can be observed that in the linear setting, the additional computational cost introduced by increasing $c$ is relatively constant. Even for the smallest MLP, it accounts for only approximately 3% of the total cost and can therefore be neglected.

In the nonlinear case, we measured the time required to compute $c$ under $K_{\text{few}} = 1, 2$, and 5 Newton iterations. The Newton updates are performed only during the first $1/6$ of the training process. Once the error stabilizes, the value of $c$ is fixed. The results are reported in Table 16.

*Table 16.* The computational cost of CAML in non-linear PDEs. FP: the forward propagation process; AD: automatic differentiation of PDE residual and boundary conditions; AC: calculation of the additive constant $c$; BP: the backward propagation process; Percentage: The proportion of time spent on calculating $c$ in the total duration.

|  |  |  | FP | AD | AC | BP | Percentage |
|---|---|---|---|---|---|---|---|
| $K_{\text{few}} = 1$ | MLP | Single Iteration ($ms$) | 0.1799 | 1.4494 | **1.0979** | 2.6280 | 20.50% |
|  |  | Full Process ($s$) | 1.2951 | 11.5819 | **3.9947** | 18.6597 | 11.24% |
|  | PirateNets | Single Iteration ($ms$) | 0.7328 | 8.0003 | **1.0322** | 14.8213 | 4.20% |
|  |  | Full Process ($s$) | 6.0822 | 55.2902 | **4.0421** | 112.3832 | 2.27% |
|  | PINNsFormer | Single Iteration ($ms$) | 2.5648 | 23.8952 | **1.8702** | 55.1467 | 2.24% |
|  |  | Full Process ($s$) | 7.2837 | 70.0029 | **0.6324** | 151.3379 | 0.28% |
| $K_{\text{few}} = 2$ | MLP | Single Iteration ($ms$) | 0.1895 | 1.4653 | **2.0220** | 2.7328 | 31.55% |
|  |  | Full Process ($s$) | 1.4321 | 12.8582 | **8.3121** | 21.5747 | 18.81% |
|  | PirateNets | Single Iteration ($ms$) | 0.7807 | 7.2948 | **2.3412** | 15.0520 | 9.19% |
|  |  | Full Process ($s$) | 6.1321 | 59.8055 | **9.3241** | 116.2348 | 4.87% |
|  | PINNsFormer | Single Iteration ($ms$) | 2.3818 | 21.5656 | **3.0841** | 47.1944 | 4.16% |
|  |  | Full Process ($s$) | 7.8837 | 71.2378 | **1.0790** | 155.3432 | 0.46% |
| $K_{\text{few}} = 5$ | MLP | Single Iteration ($ms$) | 0.1614 | 1.5289 | **5.0107** | 2.4561 | 54.72% |
|  |  | Full Process ($s$) | 1.3320 | 10.0022 | **19.7822** | 16.9772 | 41.13% |
|  | PirateNets | Single Iteration ($ms$) | 0.7361 | 8.2706 | **5.2069** | 15.1342 | 17.74% |
|  |  | Full Process ($s$) | 6.9929 | 57.9483 | **21.3250** | 118.1247 | 10.43% |
|  | PINNsFormer | Single Iteration ($ms$) | 2.3477 | 22.1690 | **6.0726** | 49.3347 | 7.60% |
|  |  | Full Process ($s$) | 6.4899 | 73.5383 | **2.7600** | 157.1937 | 1.15% |

The results indicate that, for nonlinear PDEs, under the typical CAML configuration ($K_{\text{few}} = 2$), the cost of online update $c$ accounts for approximately 20% of the total training time in lightweight models such as MLP. Since the computational cost of Newton update remains relatively constant across different backbones, this proportion decreases to below 5% for more complex models. Moreover, if the updates of $c$ were terminated earlier, this proportion would be further reduced.

## K. Difference with Learnable Bias

In this section, we perform an experiment to evaluate the distinction between CAML and explicitly introduced learnable biases in the output layer. Using the Heat benchmark as an illustrative example, we employ the MLP backbone and keep all experimental settings consistent with those in the main experiments. We compare the following configurations: *(i)* standard PINN with a standard MLP, *(ii)* standard PINN with an MLP augmented by an explicitly added, learnable output bias, and *(iii)* CAML with aligned constraints only (to isolate the effect of the delay-residual). The results are shown in Table 17.

*Table 17.* Comparison of CAML with a learnable output bias.

|  | Stp | $L_2$ | $\cos(\phi)$ |
|---|---|---|---|
| PINN | $10127_{\pm 2839}$ | $7.52\text{e-}3_{\pm 4\text{e-}3}$ | 5.89% |
| PINN + Learnable Bias | $9264_{\pm 1877}$ | $6.53\text{e-}3_{\pm 2\text{e-}3}$ | 16.13% |
| CAML (AC-Only) | $1491_{\pm 186}$ | $1.15\text{e-}3_{\pm 6\text{e-}5}$ | 36.12% |

As shown in Table 17, introducing a bias term improves over vanilla PINN in all metrics, indicating that increasing the constant-mode flexibility indeed alleviates part of the gradient conflict. However, the improvement remains limited: convergence is still slow, and the final error remains an order of magnitude larger than that of CAML. In contrast, CAML (AC-only) significantly reduces the required training steps, lowers the $L_2$ error, and substantially increases the gradient alignment ratio. This suggests that simply introducing a trainable bias is insufficient. The key advantage of CAML lies in its per-iteration optimal alignment of the constant mode via an analytic (or low-dimensional) solve, rather than relying on gradient-based adaptation of a bias parameter, which may become trapped in a distorted residual valley.

## L. Parameter Sensitivity

In this section, we analyze the sensitivity of CAML to its key hyperparameters, including the Newton iteration steps $K_{\text{init}}$ and $K_{\text{few}}$, the constant-fixing iteration $t_c$, and the delay-residual parameters $t_d$ and $t_r$.

**Newton's method.** The value of $K_{\text{init}}$ is set higher because, in the first step, $c$ does not have a warm start, and more iterations are needed to find reasonable initial values. $K_{\text{few}}$ is set to 2 as a trade-off between accuracy and computational cost (the comparison of costs is shown in Table 15). The selection of $t_c$ is based on observing when the change in $c$ stabilizes, thereby avoiding training noise caused by minor fluctuations in the later stage.

**Delay-residual schedule.** The principle for setting $t_d$ and $t_r$ is to ensure that the network roughly satisfies the boundary conditions before the PDE residual term is introduced. To examine the effect of these parameters more systematically, we conducted an additional experiment on the Poisson benchmark with the MLP backbone (5 random seeds). The results are reported in Table 18.

*Table 18.* Sensitivity of CAML to the delay-residual schedule $(t_d, t_r)$ on the Poisson benchmark (MLP backbone). The definitions of Stp and $L_2$ are consistent with those in the main experiment. Configurations that cannot achieve the expected accuracy within the maximum training budget $T_{\max}$ are indicated by '$-$'.

| $t_d/t_r$ | 0/0 | 40/160 | 80/320 | 120/480 | 160/640 | 200/800 | 800/3200 |
|---|---|---|---|---|---|---|---|
| Stp | $-$ | 5521 | 4713 | 5595 | 4574 | 3868 | 1984 |
| $L_2$ | $-$ | 9.99e-3 | 6.25e-3 | 1.00e-2 | 6.98e-3 | 5.00e-3 | 2.98e-3 |

Overall, we found that for strongly nonlinear boundary conditions, the later the residuals are introduced, the more stable the training becomes. Nevertheless, excessive $t_d$ may lead to overfitting of the boundary conditions, thereby causing gradient explosion after the introduction of the PDE residual term. Therefore, we suggest setting this value with greater caution.

## M. Failure Experiment

In this section, we present a failure-case experiment as a consistency check of our theoretical analysis. We consider a well-posed toy Poisson problem with purely Dirichlet boundary conditions. In this case, the model's initialization is already close to the global optimum, and the PDE loss landscape is relatively smooth, so it will not guide the parameter to a remote valley at the very beginning. In this setting, CAML is not expected to provide a significant improvement over standard PINNs. This experiment clarifies the applicability and limitations of our method.

Consider the Poisson equation on a unit square domain $\Omega = [0,1] \times [0,1]$:

$$-\nabla^2 u(x,y) = f(x,y), \quad (x,y) \in \Omega, \tag{101}$$

where $u(x,y)$ is the unknown solution and $f(x,y)$ is the source term. For this case, the source term is chosen as:

$$f(x,y) = -2\pi^2 \sin(\pi x) \sin(\pi y). \tag{102}$$

The boundary condition is specified as:

$$u(x,y) = 0, \quad (x,y) \in \Gamma, \tag{103}$$

and the analytical solution for this problem is:

$$u(x,y) = \sin(\pi x) \sin(\pi y). \tag{104}$$

In this experiment, we measured the performance of the MLP backbone under the original PINN and CAML (AC-only). We simultaneously recorded the final additive offset $c$ applied by CAML.

*Table 19.* Comparison of CAML on toy Poisson benchmark.

|  | Stp | $L_2$ | $\cos(\phi)$ | $c$ |
|---|---|---|---|---|
| PINN | $2472_{\pm 398}$ | $3.96\text{e-}3_{\pm 9\text{e-}4}$ | 19.88% | — |
| CAML (AC-Only) | $2328_{\pm 221}$ | $3.92\text{e-}3_{\pm 6\text{e-}4}$ | 23.21% | 3.16 |

Based on the results in Table 19, the toy Poisson experiment indicates that for well-posed problems with purely Dirichlet boundary conditions, CAML (AC-only) does not provide substantial improvements over standard PINNs. Both methods achieve nearly identical convergence steps and final $L_2$ errors, with only a modest increase in gradient cosine similarity under CAML. The learned additive offset $c$ remains small, yielding only minor performance gains. This result is consistent with our theoretical discussion: when initialization has already eliminated additive invariance and the optimization landscape is relatively benign, the optimizer's search distance within the PDE residual loss valley is already short. Therefore, the advantage of solution set expansion caused by aligned constraints becomes limited.

## N. Benchmark Visualization

In this section, we present visualization results to qualitatively assess the performance of CAML integrated into the different backbones. We visualize snapshots of the solution on a two-dimensional domain across different benchmarks.

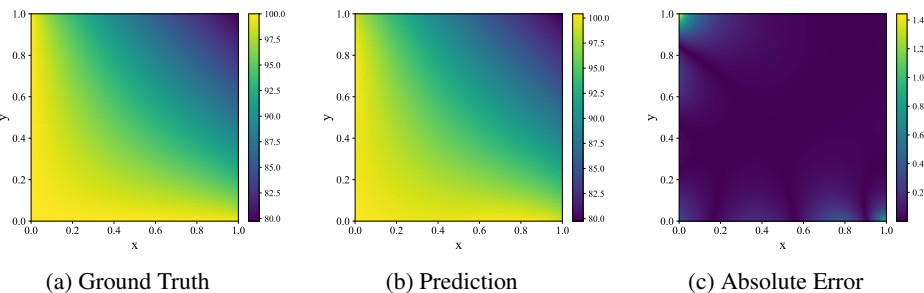

(a) Ground Truth      (b) Prediction      (c) Absolute Error

*Figure 10.* Visualization results on the Heat benchmark, showing the ground truth, prediction and error distributions during training with CAML integrated into the MLP backbone.

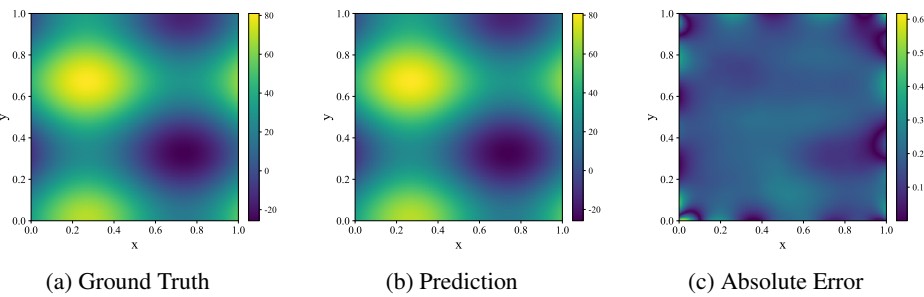

(a) Ground Truth        (b) Prediction        (c) Absolute Error

*Figure 11.* Visualization results on the Poisson benchmark, showing the ground truth, prediction and error distributions during training with CAML integrated into the MLP backbone.

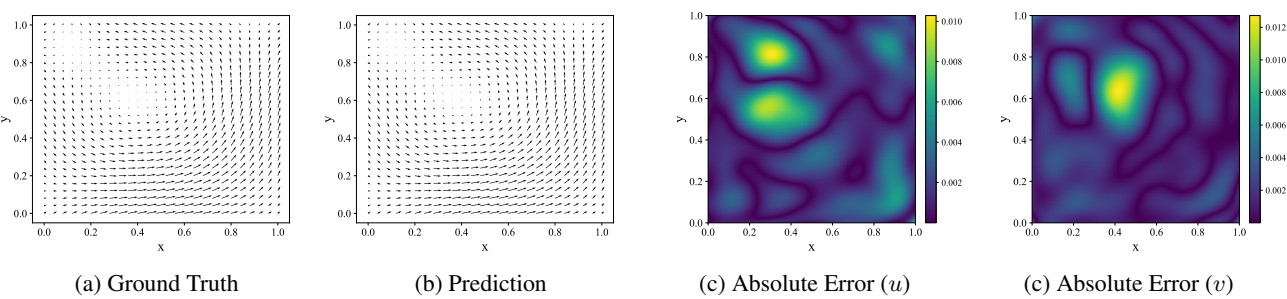

(a) Ground Truth     (b) Prediction     (c) Absolute Error ($u$)     (c) Absolute Error ($v$)

*Figure 12.* Visualization results on the NS benchmark, showing the ground truth, prediction and error distributions during training with CAML integrated into the MLP backbone.

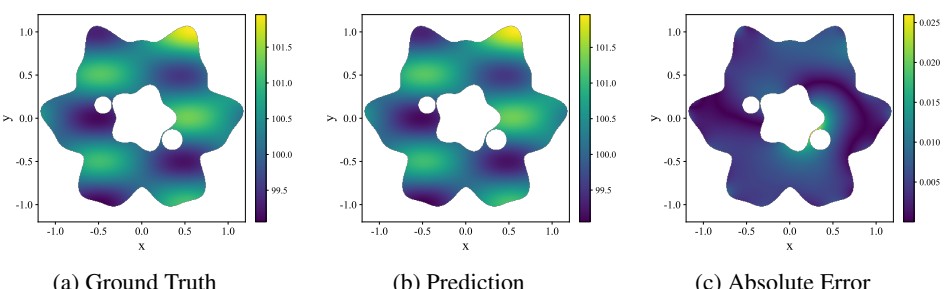

(a) Ground Truth        (b) Prediction        (c) Absolute Error

*Figure 13.* Visualization results on the Helmholtz benchmark, showing the ground truth, prediction and error distributions during training with CAML integrated into the MLP backbone.

