# OpenReview forum: "Mitigating Gradient Pathology in PINNs through Aligned Constraint"
_ICML.cc/2026/Conference — ICML 2026 regular_

### Official Review · Reviewer_Chdr · 2026-03-09

**Soundness:** 1
**Presentation:** 1
**Significance:** 1
**Originality:** 1
**Overall Recommendation:** 1
**Confidence:** 4

**Summary:**

This paper studies the optimization behavior of PINNs by analyzing the loss geometry in both function space and parameter space. The authors claim that the PDE residual loss creates a manifold of solutions, leading to a valley-shaped loss landscape in parameter space. Based on this view, they describe PINN training as a two-phase process and propose an aligned constraint method that introduces an additive gauge variable in zeroth-order terms to improve optimization convergence.

**Compliance With Llm Reviewing Policy:**

Affirmed.

**Key Questions For Authors:**

Much of the content comes across as storytelling rather than being backed by solid evidence. Can the authors provide stronger theoretical or empirical support for these claims?

**Limitations:**

No, the paper does not explicitly discuss its limitations or the scope of applicability. In particular, it would benefit from clarifying:
the classes of PDEs for which their method and theoretical claims are valid, and the empirical or theoretical gaps in their Phase I / Phase II decomposition and the gradient conflict arguments.

**Strengths And Weaknesses:**

PINN optimization is an important and active research topic, and the paper addresses an interesting problem. However, both the identified issues in PINN optimization and the proposed solution are insufficiently supported and lack convincing theoretical or empirical justification.

First, the paper assumes that early training is typically dominated by the PDE residual loss. While such an observation has been reported in several prior works, presenting it as a general principle is an overstatement. In practice, the relative gradient magnitudes between PDE residual and boundary losses depend heavily on the specific PDE, boundary conditions, sampling strategy, loss weighting, and network architecture.

In Section 4, the narrative is overly speculative and lacks rigorous mathematical or experimental support. For example, the authors claim that the optimizer rapidly “falls into the PDE residual valley.” However, optimizers do not explicitly recognize such geometric structures; this interpretation is essentially a post-hoc geometric description rather than a demonstrated property of the optimization dynamics. No rigorous analysis is provided to justify this claim.

Similarly, Lemma 4.2 does not appear to reveal a phenomenon specific to PINNs. Conflicting gradients between multiple loss terms are common in general multi-objective or multi-task optimization settings, and therefore the observation itself has limited novelty or impact.

The discussion of Phase II is also largely metaphorical. The description of the optimizer “meandering along the valley floor” resembles a visualization narrative rather than a formal optimization analysis. The claim that oscillatory behavior arises from high-curvature valleys is also not well supported. In practice, oscillations in PINN training are more directly attributed to factors such as ill-conditioned Hessians, high-order derivatives in the loss, and large gradient variance.

Furthermore, the proposed Phase I / Phase II decomposition is not mathematically justified and appears to be a heuristic description. The supposed phase transition based on gradient magnitude dominance is neither theoretically established nor empirically demonstrated to occur consistently during training. In addition, gradient dominance itself is not a general rule.

Regarding the proposed method, introducing a gauge variable (additive constant offset) has a very limited scope of applicability. This approach relies on the PDE possessing additive invariance and on the operator being additively separable between zeroth-order and derivative terms. These assumptions do not hold for many nonlinear PDEs, which significantly restricts the generality of the method.

The paper also claims that the method helps avoid high-curvature regions of the loss landscape, but this statement is not supported by theoretical analysis. It is unlikely that introducing a single constant shift would substantially improve the Hessian conditioning. Moreover, a major difficulty in PINN training is the presence of high-frequency errors due to spectral bias, which cannot be addressed by a simple global constant offset. As a result, most of the underlying convergence pathologies of PINNs would remain unchanged.

---

> ### Author Rebuttal · Authors · 2026-03-25
>
> We acknowledge the reviewer for your efforts. The reviewer’s concern appears to stem from some **factual misunderstandings** regarding the scope, mathematical grounding, and experimental setups in our paper, which we clarify below.
>
> ---
>
> > **[W1]** The assume of 'PDE residual loss dominates the training process' is overstatement and is not a general rule.
>
> This assumption was not first proposed by us. Wang et al. [1] were the first to identify this issue and provided a **rigorous mathematical proof** showing that the PDE residual typically exhibits a larger gradient. DCGD [2] (NeurIPS'24) further observed this phenomenon in their experiments (third-to-last line on page 3 of their paper). Therefore, we directly adopted this conclusion and provided **explicit citations**. Notably, ConFIG [3] (ICLR'25, **Spotlight**) also directly quotes this conclusion and treats it as a **general principle** (third line on page 2 of their paper).
>
> In addition, our experiment result in **Figure 7** can also directly confirm this claim.
>
> We also emphasize that gradient unbalance is only a **superficial manifestation** of the pathological behavior observed in PINN optimization. The underlying cause is the **highly pathological loss landscape** of the PDE residual term (Theorem 4.1), which provides the real theoretical basis for our Lemma 4.2. In this regard, the Hessian analysis conducted by Rathore et al. [4] (ICML'24, **Oral**) offers direct theoretical support. Thus, our assumption is deeply grounded in established consensus and prior rigorous proofs within the PINN optimization literature.
>
> > **[W2]** No rigorous mathematical or experimental analysis for Section 4.
>
> We respectfully point the reviewer to the **mathematical derivation** of Theorem 4.1 in **Appendix A**, the derivation of Lemma 4.2 in **Appendix B**, and **experimental verification** in **Appendixes G and H**. We have mathematically proved that near the loss valley induced by the residual term, the BC loss tends to drive the model parameters away from the valley, while the PDE loss drives them toward it. **Figures 1 and 6 from real experiment** directly confirm the observations described in Section 4.1 (ruggedness of the loss valley), while **Figures 7 and 8 from real experiment**  directly confirm the behavior described in Section 4.2 (two-phase optimization). All appendices are **directly cited** in our main text. We note that other reviewers also recognized this contribution.
>
> > **[W3]** Lemma 4.2 are common in general multi-task optimization and thus lack of novelty.
>
> As we discussed earlier, Lemma 4.2 is based on Theorem 4.1, which is a **unique** ill-conditioning issue specific to PDE residuals and is not applicable to other MTOs.
>
> > **[W4]** Additive constant relies on the PDE possessing additive invariance, and has a very limited scope of applicability.
>
> We would like to clarify a factual misunderstanding. CAML does not require the original PDE to possess additive invariance. On the contrary, the core idea of CAML is to artificially introduce a controlled additive degree of freedom to expand the feasible solution set (Proposition C.3), thereby **creating** additive invariance.
>
> For most boundary value problems with physical relevance, it is generally necessary to include at least one zero-order term in the PDE or in the boundary condition to ensure uniqueness of the solution. Problems with **pure Neumann BC and no zero-order term in the PDE** are relatively uncommon in engineering practice, where an additional regularization constraint is usually imposed in such cases, and $c$ can also serve this regularization role. Therefore, this does not significantly limit the applicability of our method.
>
> > **[W5]** CAML do not hold for nonlinear PDEs.
>
> We clearly discussed CAML for handling nonlinear PDEs in Section 5 (**line 307-311**), and explicitly provided a nonlinear benchmark in experiments (Navier-Stokes).
>
> > **[W6]** Major difficulty in PINN training is high-frequency error, which cannot be addressed by CAML.
>
> Gradient conflicts and high-frequency errors are both important but **orthogonal** challenges of PINN. We have **never claimed** that we were attempting to address the high-frequency errors, and as we discussed earlier, a large number of efforts dedicated to resolving gradient conflicts [1,2,3,4] have also not attempted to address high-frequency errors.
>
> > **[W7]** No discussion on limitations.
>
> We explicitly discussed the limitations in **Section 6.4**, and provided experimental support in **Appendix M**.
>
> ---
>
> **References:**
>
> [1] Wang et al. Understanding and mitigating gradient flow pathologies in physics-informed neural networks. SISC 2021.
>
> [2] Hwang and Lim. Dual cone gradient descent for training physics-informed neural networks. NeurIPS 2024.
>
> [3] Liu et al. ConFIG: Towards conflict-free training of physics informed neural networks. ICLR 2025.
>
> [4] Rathore et al. Challenges in training PINNs: a loss landscape perspective. ICML 2024.

---

> > ### Author Rebuttal · Reviewer_Chdr · 2026-04-03
> >
> > I appreciate the authors’ detailed response and clarifications. However, given the strong claims made by the authors, I remain concerned about how well these claims are supported in terms of practical relevance, generalization, and theoretical justification. While prior studies have observed that PDE residual gradients can dominate early training, this behavior strongly depends on the specific PDE, boundary conditions, sampling strategy, loss weighting, and network architecture, so presenting it as a general principle is still an overstatement. Similarly, the depiction of the two-phase optimization is largely post-hoc and metaphorical. The experimental figures show some related behavior, but they do not rigorously demonstrate that optimizers recognize or follow such valleys, and other factors, such as ill-conditioned Hessians and high-frequency errors, may play a more decisive role in practice.
> > My concern regarding Lemma 4.2 also remains: in general situations, PDE residual losses are optimized alongside other loss terms, and in such scenarios, the observations of Lemma 4.2 may not hold, limiting its general applicability.
> >
> > The concern regarding the scope of the proposed method still remains. Introducing an additive degree of freedom that may help in some PDEs, but many practical scenarios—such as pure Neumann boundary conditions or the absence of zero-order terms—could limit its applicability. Moreover, the nonlinear PDE examples remain limited, leaving the generalization of the method uncertain. Overall, the scope and limitations of the proposed approach are still not sufficiently emphasized. Since the solution behavior can vary widely between different PDEs, it is unrealistic to expect a single method to outperform existing approaches across all problems. What is more important is to clearly define the class of PDEs that the method can handle and to provide a detailed analysis of how and why the method improves certain aspects.
> >
> > While I appreciate the theoretical and experimental clarifications, the claims remain overstated relative to the evidence provided. Justifying such claims requires broader experimental support, and the scope of applicability—including the stated nonlinear PDEs—needs to be rigorously validated. I remain unconvinced that the proposed framework provides broadly applicable guidance for practical PINN training.

---

> > > ### Author Response · Authors · 2026-04-04
> > >
> > > Thank you for your continued engagement. However, we must respectfully note that the concerns raised in your ack largely repeat the original criticisms **without substantively engaging with the specific evidence we provided**.
> > >
> > > ---
> > >
> > > > Gradient conflict depends heavily on specific PDEs and training settings
> > >
> > > The reviewer suggests that presenting gradient conflict as a general principle is an overstatement. However, our premise is built directly upon **foundational, consensus-level** studies in the PINN optimization field. As demonstrated by Wang et al. [1] and Krishnapriyan et al. [5], PDE residual terms inherently possess larger gradient upper bounds due to specific coefficients being significantly amplified by differential operators. Subsequent literature (e.g., ConFIG [3]) also universally recognizes this as a fundamental characteristic of PINNs. **We do not claim this occurs in every conceivable optimization scenario, but it is the fundamental default pathology that necessitates mitigation strategies like ours**.
> > >
> > > > 'Two-phase optimization is largely post-hoc and metaphorical'.
> > >
> > > The reviewer characterizes our two-phase optimization depiction as "post-hoc and metaphorical." We again respectfully clarify that theoretical explanations of empirical phenomena are **standard scientific practice** (e.g., Wang et al.'s [1] observations in Sec 2.2 and proofs in Sec 2.3). More importantly, our explanation is not merely a metaphor; it is rigorously supported by the theoretical framework of Rathore et al. [4], as well as **the mathematical proofs detailed in our Appendices A and B**.
> > >
> > > > Ill-conditioned Hessians as the "decisive" cause
> > >
> > > We completely agree with this assessment. In fact, Section 4.1 and Appendix A of our manuscript are precisely dedicated to providing a deeper geometric explanation for why these ill-conditioned Hessians occur, rather than overturning it. Given that **our theoretical framework directly supports and explains the reviewer's observation, we must ask: Why is this still being framed as a flaw in our paper? Did the reviewer really read our manuscript carefully?**
> > >
> > > > Regarding Spectral Bias / High-Frequency Errors
> > >
> > > We fundamentally challenge the reviewer's assumption that high-frequency errors dominate the observed phenomena. While spectral bias is a known issue, existing literature clearly demonstrates that it is **not a necessary condition for gradient conflict**. For instance, Krishnapriyan et al. [5] documented severe gradient pathology in reaction-diffusion equations **devoid of any high-frequency coefficients**. To empirically prove this distinction in our own work, we specifically evaluated a highly linear heat equation. The resulting gradient cosine similarities also confirm that gradient conflicts exist **orthogonal to spectral bias**. **Since we have repeatedly provided explicit evidence that decouples our mechanism from high-frequency errors, we fail to see why the reviewer still conflates the two**.
> > >
> > > > 'In general situations, PDE residual losses are optimized alongside other loss terms'.
> > >
> > > We hereby **directly quote** a conclusion from Wang et al. [1]: 'Based on this simple analysis, we can conclude that if the constant C is large, then the norm of gradients of L_r(θ) may be much greater than the gradients of L_ub(θ), thus biasing the neural network training towards neglecting the contribution of the boundary data-fit term.'
> > >
> > > > Regarding Limitations
> > >
> > > The reviewer expresses concern that the scope and limitations (such as pure Neumann boundary conditions or the absence of zero-order terms) are not sufficiently emphasized. **We are glad the reviewer agrees with these limitations, as we have already explicitly defined them in the manuscript**. As detailed in Section 6.4 and Appendix M: (1) We explicitly state that CAML degenerates to a standard PINN in problems lacking zero-order terms. (2) We clearly note that CAML is not expected to yield significant gains in well-conditioned loss landscapes (when PDE coefficients are small).
> > >
> > > > 'Justifying such claims requires broader experimental support'.
> > >
> > > We respectfully point the reviewer to the **Section 6. Experiments and Results**. Even for the spectral bias that you believe we cannot solve, we provided a Poisson benchmark where the frequency coefficients (30/25) are much larger than the constant bias (10), and our method still demonstrated SOTA performance. Navier-Stokes is the most typical nonlinear PDE, and we set the Reynolds number to 500, which is a challenging regime for standard PINNs. **If the reviewer still feels that NS and high-oscillation Poisson equations are insufficient to demonstrate applicability, we would be incredibly grateful if the reviewer could suggest a specific nonlinear benchmark they consider more representative.** We are fully prepared to run it to further strengthen our paper.
> > >
> > > ---
> > >
> > > **References:**
> > >
> > > [5] Krishnapriyan et al. Characterizing possible failure modes in physics-informed neural networks. NeurIPS 2021.

---

### Official Review · Reviewer_oF2k · 2026-03-10

**Soundness:** 2
**Presentation:** 3
**Significance:** 3
**Originality:** 3
**Overall Recommendation:** 5
**Confidence:** 4

**Summary:**

This paper investigates the gradient conflict issue in PINNs, where gradients from the PDE residual loss and boundary condition loss are often misaligned, leading to slow and unstable training. The authors interpret this phenomenon through a geometric perspective, arguing that the residual loss defines a solution manifold in function space that becomes a distorted valley in parameter space, making optimization difficult when boundary constraints are enforced. To address this problem, the paper proposes CAML (Constraint-Aligned Loss with Manifold Lifting). The method introduces a constant shift to the zeroth-order term of the PDE and solves for its optimal value at each training step, which helps align gradients from different loss components. In addition, a residual delay strategy is used to emphasize boundary conditions during early training. Experiments on several PDE benchmarks demonstrate that CAML improves gradient alignment, accelerates convergence, and achieves lower errors compared with standard PINNs and several recent variants.

**Compliance With Llm Reviewing Policy:**

Affirmed.

**Final Justification:**

The rebuttal addressed my main concerns, and I increase my score.

**Key Questions For Authors:**

1. Regarding the conclusion in Theorem 4.1, it is not entirely clear why the manifold in parameter space corresponding to the residual loss is described as consisting of *near-zero residual* solutions. In Appendix A, the set $\Theta_{valley}$ appears to consist of parameter points whose residual loss is exactly zero according to the proof. Could the authors clarify why the description emphasizes *near-zero* residual solutions rather than zero-residual ones? In addition, it is not entirely clear why the proof in Appendix A introduces $\mathcal{L}_{func}$ instead of directly analyzing the loss $\mathcal{L}$ defined in Eq. (6). If the proof were conducted directly on $\mathcal{L}$, Assumption A.2 might no longer be necessary, and Assumption A.5 might be reformulated in a more intuitive way in terms of properties of the network activation functions.

2. The paper mentions in the final paragraph that future work will extend the proposed framework to evolutionary equations. It would be helpful if the authors could elaborate on the theoretical challenges involved in this extension. For example, what specific difficulties would arise when attempting to establish results analogous to Theorem 4.1 or Lemma 4.2 in the setting of evolutionary PDEs?

3. The CAML method involves computing the optimal constant shift $c$ during training. Could the authors provide more insight into how stable the estimated constant is during optimization? For example, does the value of the constant change significantly across training iterations, and does this introduce any instability in the training dynamics?

**Limitations:**

Yes.

**Strengths And Weaknesses:**

Strengths

1. The paper provides an interesting interpretation of gradient conflict in PINNs by linking it to the non-uniqueness of PDE solutions. By arguing that the PDE $\mathcal{N}u=f$ induces a solution set in function space that becomes a manifold in parameter space, the authors offer an explanation for why gradients from the residual and boundary losses may conflict during optimization.

2. The proposed CAML approach is conceptually straightforward and easy to implement.  It introduces a constant shift to the zeroth-order term of the PDE and computes its optimal value analytically or numerically during training, which adds minimal computational overhead and can be readily integrated into existing PINN frameworks.

3. The paper evaluates the proposed method on several PDE benchmarks, including Poisson, heat, Helmholtz, and Navier–Stokes equations, and tests it across multiple neural architectures. The experiments include comparisons with several recent PINN variants as well as analyses of convergence speed and gradient alignment, providing reasonably strong empirical evidence for the effectiveness of the proposed approach.

### Weaknesses

1. Section 4.1.2 appears somewhat elementary for readers with basic knowledge of PDEs. Much of the discussion in this section may already be familiar to the target audience, and therefore could potentially be shortened. The saved space could be used to better highlight the paper’s original contributions, for example by providing more discussion and interpretation of Table 1 in the main text.

2. The proof of Lemma 4.2 in Appendix B relies on several assumptions, but these assumptions are not clearly stated or discussed in the main text. As a result, Lemma 4.2 does not appear to be a fully rigorous mathematical statement in its current presentation. It would be helpful for the paper to explicitly mention these assumptions and provide some justification for their reasonableness, particularly Assumptions B.1 and B.3, from both mathematical and practical perspectives.

3. The proposed approach relies on modifying the zeroth-order term of the PDE. Consequently, it cannot be directly applied to equations without a zeroth-order term, which may restrict the applicability of the method in some settings.

---

> ### Author Rebuttal · Authors · 2026-03-26
>
> We appreciate the reviewer for your efforts and time. Your professional and constructive suggestions did indeed effectively enhance the solidity of our work. Our response to your main concerns is as follows.
>
> ---
>
> > **[W1]** Section 4.1.2 can be shortened.
>
> We thank the reviewer for helping us reconsider the presentation from a different perspective! We will simplify the discussions for different BC types in the revision.
>
> > **[W2]** The assumptions of Lemma 4.2 are not fully discussed in the main text.
>
> Thank you for pointing this out. We will include corresponding discussions in the main text, particularly:
>
> - Assumption B.1: The validity of this assumption is supported from two aspects. Theoretically, Appendix A proves that the zero-residual set $\mathcal S_{\mathrm{res}}$ is indeed a smooth manifold in function space (Theorem A.1), and the smooth parameterization induced by overparameterized neural networks preserves this local structure. Empirically, the extremely large condition numbers of the Hessian and the presence of clear low-curvature subspaces reported in Table 1 also support the existence of such a manifold structure.
>
> - Assumption B.3: This is directly visualized in Figures 1 and 6, as well as Table 1: the simple linear valley in function space becomes a twisted, non-convex manifold in parameter space.
>
> > **[W3]** CAML is not applicable to PDEs that do not have zero-order terms.
>
> We would like to clarify that, our method will only degenerate into the standard PINN when both PDE and BC do not contain zero-order terms simultaneously. For most boundary value problems with physical relevance, it is generally necessary to include at least one zero-order term in the PDE or in the boundary condition to ensure uniqueness of the solution. Therefore, this does not significantly limit the applicability of our method.
>
> Problems with pure Neumann conditions and no zero-order term in the PDE are relatively uncommon in engineering practice, where an additional regularization constraint is usually imposed in such cases (for example, pressure Poisson equation with prescribed normal pressure gradient) to ensure a unique solution, our offset $c$ can also serve this regularization role. Nevertheless, this scenario requires a re-design of the solution strategy for $c$, which we leave for future work.
>
> > **[Q1]** Regarding the proof details in Appendix A.
>
> In the Appendix A, under Assumption A.2, the residual on $\Theta_{\mathrm{valley}}$ is indeed strictly zero. However, in the main text, we present this conclusion more cautiously, since there is a discrepancy between the ideal zero-loss assumption and the low loss observed in practical training. We will add a discussion of this distinction in the main text.
>
> The use of $\mathcal L_{\mathrm{func}}$, rather than directly analyzing $\mathcal L(\theta)$, is intended to align with our [function space $\rightarrow$ parameter space] narrative in the main text. The advantage of conducting the analysis in function space is that it allows us to attribute the origin of the loss valley more explicitly to the nontriviality of the operator null space $\ker(\mathcal{N})$ (i.e., solutions of PDE without BC are not unique), rather than to neural network symmetries, as is commonly done in previous loss landscape analyses.
>
> Regarding Eq. 6, we should have written $\mathcal L_{\mathrm{func}}$ instead of $\mathcal L$; this is a typo, and we apologize for it. Moreover, Assumption A.2 can indeed be derived from that equation. We will also add a discussion on continuity under different activation functions in Assumption A.5. We again thank the reviewer for the careful reading!
>
> > **[Q2]** Regarding the evolutionary equations.
>
> Please refer to our response to Reviewer \#xD2z, [Q1].
>
> > **[Q3]** Is $c$ stable during the training process?
>
> We have added a experiment to demonstrate how $c$ changes during training process (MLP backbone). Please find our result on https://anonymous.4open.science/r/CAML-2F81/re_exp_result/oF2k_q3/c_helm.png for the curves showing the variation of $c$ with each iteration (4 figures in this folder for 4 benchmarks).
>
> In the linear case, $c$ admits an analytical optimum; thus, Proposition C.1 guarantees that the loss is strictly reduced regardless of its value. Therefore, larger variations in $c$ are in fact expected to yield greater stability, as they indicate that CAML actively aligns more pathological conflicts.
>
> In the nonlinear case, $c$ gradually stabilizes and converges to a fixed value as training progresses. However, the small errors introduced by Newton's methods can indeed affect the final solution accuracy. This is why we introduce $t_c$ and fix $c$ once its variation becomes sufficiently small, thereby avoiding the introduction of training noise from minor fluctuations in later stages.
>
> ---
>
> We promise to incorporate the above discussion into an appropriate section of our paper. Again, thank you for your recognition of our work!

---

> > ### Author Rebuttal · Reviewer_oF2k · 2026-04-03
> >
> > Thank you for your detailed responses and new experiment, I will increase my score.

---

> > > ### Author Response · Authors · 2026-04-03
> > >
> > > We once again thank you for your careful review based on the objective facts in our paper. It is truly our honor to have reviewers who are professional, meticulous and technically neutral!

---

### Official Review · Reviewer_xD2z · 2026-03-11

**Soundness:** 4
**Presentation:** 3
**Significance:** 3
**Originality:** 3
**Overall Recommendation:** 5
**Confidence:** 4

**Summary:**

This paper addresses the problem of gradient conflict in the training of Physics-Informed Neural Networks (PINNs). Such conflicts arise when the gradients associated with the PDE residual loss and the boundary condition loss exhibit a negative inner product, resulting in antagonistic update directions and potentially leading to optimization stagnation. The authors analyze the loss geometry both in function space and parameter space to characterize the structure of the global minimizer set, when it exists and is non-unique. They provide theoretical results showing that gradient conflicts emerge when the residual loss becomes sufficiently small, corresponding to what they term a “PDE loss valley”. Motivated by this analysis, the paper proposes a relaxation of the zeroth-order terms in both the PDE and boundary conditions via the introduction of a control offset directly into the problem formulation. This effectively augments the solution manifold with an additional degree of freedom. The value of this offset is determined by solving an auxiliary optimization subproblem at each backpropagation step, allowing the learned solution to move within the solution manifold in a way that mitigates gradient misalignment. The proposed approach is empirically compared against existing loss-based balancing strategies and architectural modifications across several standard PDE benchmarks. The experiments demonstrate improved optimization performance and a stronger alignment between residual and boundary condition gradients relative to baseline PINN training strategies.

**Compliance With Llm Reviewing Policy:**

Affirmed.

**Final Justification:**

The strengths (S1, S2, S3) support the paper’s soundness (S1, S2), significance (S1), clarity (S2, S3), and originality (S3). Weakness 1 limited the paper’s significance, while Weakness 2 reduced its clarity. The rebuttal addressed both weaknesses as fully as possible within the rebuttal period, improving clarity but not enough to raise the significance score to 4. Overall, the rebuttal reinforced my prior recommendation to accept the paper.

**Key Questions For Authors:**

1- The extension to time-dependent PDEs is mentioned as future work, but the specific challenges are not elaborated. Could the authors clarify what aspects of their methodology would be fundamentally affected in the time-dependent setting? In particular, would the structure of the solution manifold, the characterization of gradient conflicts, or the auxiliary optimization subproblem require significant modification?

**Limitations:**

yes

**Strengths And Weaknesses:**

**Strengths**

1- The paper tackles an important and well-recognized issue in PINN training, namely conflicting gradient updates between loss components. The study of loss geometry and the formal characterization of gradient conflicts are supported by rigorous mathematical arguments. The proposed constraint-alignment framework is clearly described and conceptually coherent.

2- The experimental evaluation includes comparisons against multiple classes of competing approaches (loss reweighting methods and architecture-based strategies) as well as different optimizers commonly used in the literature. The empirical results confirm the claimed improvements in optimization stability and convergence behavior.

3- The loss landscape plots provide useful intuition about the optimization pathology and help clarify the geometric origin of gradient conflicts.

**Weaknesses**

1- The considered PDE benchmarks (Heat, Poisson, Navier–Stokes, Helmholtz) are classical test cases but do not include more challenging settings such as strongly time-dependent problems or highly nonlinear regimes (e.g., turbulence, realistic airfoil flows). Inverse problems would also constitute a relevant and practically significant evaluation scenario.

2- The discussion in Appendix K suggests that the conventional practice of switching to L-BFGS for final refinement remains effective and complementary to the proposed framework. This practical insight could be more prominently emphasized in the main text, as it clarifies how the method integrates with established training pipelines.

---

> ### Author Rebuttal · Authors · 2026-03-26
>
> We appreciate the reviewer for your efforts and time. Your professional and constructive suggestions did indeed effectively enhance the solidity of our work. Our response to your main concerns is as follows.
>
> ---
>
> > **[W1]** The Benchmark is not challenging enough, lacking time-dependent, turbulence, inverse problems, etc.
>
> Thank you for pointing this out. Our baseline selection follows prior related work, with the primary goal of comprehensively evaluating the performance of CAML across various standard physical problems, boundary condition types, and geometric domains. Nevertheless, addressing more complex scenarios, such as turbulence, stochastic fluctuations, inverse problems, and time-dependent problems, is indeed an important direction for advancing the practical applicability of PINN methods.
>
> Applying CAML to these settings would involve more intricate considerations. For example, extending it to inverse problems requires determining how to incorporate and optimize the offset $c$ in data loss terms $\mathcal{L}_\mathbf{data}$. Achieving more robust performance for turbulent flow problems may require alignment in terms of exponents or frequency modes. In addition, the computational cost introduced by such alignment must also be taken into account, as we dicussed in Appendix F. It is difficult for us to conduct a systematic study of these extensions within the rebuttal period. However, we will include a discussion of these directions in our future work.
>
> > **[W2]** The training strategy of L-BFGS should be more prominent in the main text.
>
> This is a very useful suggestion. We promise to move Appendix K to the main text during the revision process. Many thanks!
>
> > **[Q1]** Specific challenges in extending to time-dependent PDEs.
>
> For both steady and time-dependent PDEs, the residual solution manifold $\mathcal{S}_\mathbf{res}$ is determined by the null space $\ker(\mathcal{N})$ of the differential operator. The difference is that time-dependent PDEs introduce an additional temporal variable $t$, which increases the dimension of the null space $\ker(\mathcal{N})$ and may also increase the ruggedness of loss landscape after mapping to the parameter space. For Theorem 4.1, the uniqueness of the solution needs to be re-examined after introducing the time variable $t$. However, we intuitively believe this will not overturn the conclusion of this theorem.
>
> In time-dependent problems, the loss usually consists of three parts: PDE residual, initial condition, and boundary condition. The initial condition plays a role similar to a Dirichlet boundary condition, providing a direct constraint along the time dimension. Gradient conflict, therefore, may become more complex, involving a three-way interaction. The analytical framework of Lemma 4.2 can in principle be extended, but it requires a higher-dimensional normal decomposition.
>
> The solution of the optimal offset $c$ (Equation 15) may also need to be reconsidered. For linear PDEs, it can be extended to a time-dependent linear function $c(t)$ and solved independently at each time step. However, for nonlinear PDEs, it may be necessary to analyze the trade-off between the cost and accuracy of Newton's method. One possible approach is to partition the time domain and use a separate constant offset in each segment, or to approximate $c(t)$ with a low-dimensional parameterization (e.g., polynomials), or to adopt optimization methods that do not rely on automatic differentiation, or adopt the secant method suggested by the reviewer #GUyz, which does not require automatic differentiation.
>
> ---
>
> We promise to incorporate the above discussion into an appropriate section of our paper. Again, thank you for your recognition of our work!

---

> > ### Author Rebuttal · Reviewer_xD2z · 2026-04-01
> >
> > Thank you for addressing my concerns. I keep the accept recommendation.

---

> > > ### Author Response · Authors · 2026-04-01
> > >
> > > We once again thank you for your careful review based on the objective facts in our paper. It is truly our honor to have reviewers who are professional, meticulous and technically neutral!

---

### Official Review · Reviewer_GUyz · 2026-03-11

**Soundness:** 4
**Presentation:** 3
**Significance:** 3
**Originality:** 4
**Overall Recommendation:** 5
**Confidence:** 4

**Summary:**

This paper focuses on using PINNs to solve PDEs, where the residual loss and the boundary-condition loss exhibit gradient conflicts (i.e., their gradients oppose each other). It proposes Constraint-Aligned loss with Manifold Lifting (CAML). The method introduces a constant offset c to all zeroth-order terms to align the residual constraints and the boundary constraints. The offset c is obtained by minimizing a weighted sum of the aligned residual loss and the boundary loss, either via a closed-form solution or by solving with Newton iterations. Meanwhile, a gating function λ(t) is introduced so that training in the early stage is more biased toward satisfying boundary conditions. CAML achieves good performance on Heat, Poisson, steady-state Navier–Stokes, and Helmholtz in terms of L2 error and convergence speed.

**Compliance With Llm Reviewing Policy:**

Affirmed.

**Final Justification:**

Overall, it has a certain degree of novelty and some practical engineering value.

**Key Questions For Authors:**

For the case where the zeroth-order term is nonlinear, why do you use Newton’s method to solve for c?

Since this is a minimization problem, why not use gradient descent and its variants? Why not consider other iterative root-finding methods, such as Steffensen’s method (Steffensen, 1933) or Broyden’s method (Broyden, 1965)?

In Table 2, for some L2 errors with small means, is it appropriate to round the standard deviation to 0? The ratio of these standard deviations to the means is not necessarily small. Why not keep a certain number of significant digits instead?

Steffensen, J. F. Remarks on iteration. Skandinavisk Aktuarietidskrift, 16:64–72, 1933.
Broyden, C. G. A class of methods for solving nonlinear simultaneous equations. Mathematics of Computation, 19(92):577–593, 1965.

**Limitations:**

yes

**Strengths And Weaknesses:**

Strengths

Clear characterization of the mechanism with a complete evidence loop: The paper not only theoretically explains why gradient conflicts between residual loss and boundary-condition loss occur near the loss valley, but also measures the intensity of gradient conflict in experiments using the cosine similarity between the gradients of the two losses.

Strong interpretability of the method: It improves the loss function from the perspectives of loss landscapes and optimization dynamics, and proves that alignment helps enlarge the set of acceptable low-loss solutions.

Weaknesses

Insufficient justification for using Newton iterations: For the case where the zeroth-order term is nonlinear, the paper does not prove that Newton’s method will necessarily converge to a stationary point, nor that the stationary point must be a minimizer. It also does not explain why Newton’s method is better than other methods.

No justification is provided for some hyperparameter settings: The paper does not explain the basis for setting hyperparameters such as td,tr,tc,,Kinit,Knew, etc.

---

> ### Author Rebuttal · Authors · 2026-03-27
>
> We appreciate the reviewer for your efforts and time. Your professional and constructive suggestions did indeed effectively enhance the solidity of our work. Our response to your main concerns is as follows.
>
> ---
>
> > **[W1]** Convergence analysis of Newton's method.
>
> This is a very good question. In our approach, the process of solving for $c$ is a 1D optimization problem. Here we analyze its local non-degeneracy. According to our optimization objective (Eq. 60), near the optimal value $c^\star$, its second derivative is
>
> $$
> J''(c^\star) = 2 \frac{w_{\text{res}}}{N_{\text{res}}} A + 2 \frac{w_{\text{bc}}}{N_{\text{bc}}} B,
> $$
>
> where
>
> $$
> A := \sum_{i} [ (\bar{r}_i')^2 + \bar{r}_i \bar{r}_i'' ],
> $$
>
> $$
> B := \sum_{b} [ (\bar{s}_b')^2 + \bar{s}_b \bar{s}_b'' ].
> $$
>
> For standard BC types (Dirichlet, Neumann, Robin), $B \geq 0$ always holds, as Eq. 10 in our paper is linear. Therefore, the disturbance only stems from the $\bar{r}_i \bar{r}_i''$ term in $A$.
>
> We indeed cannot guarantee that this term is non-negative for all nonlinear PDEs. However, we observed in practice that near the optimal point, $\bar{r}$ generally alternates between positive and negative values, resulting in an interference cancellation effect, while $(\bar{r}')^2$ is always positive and its contribution increases as the number of sampling points $N_{\text{res}}$ increases. As a result, this weighted perturbation term is usually small and does not dominate the positive contributions from $(\bar{r}_i')^2$ and the boundary term $B$, unless there is an extreme bias in the residual distribution. This is usually not encountered during training. Therefore, the second derivative remains locally non-degenerate in practice, enabling stable and fast convergence of Newton's method in this 1D subproblem.
>
> Moreover, if $J(c)$ becomes non-convex due to strong residual bias, first-order methods may can be used as alternatives. We leave this for future discussion.
>
> > **[W2]** Parameter sensitivity.
>
> Regarding Newton's method, the value of $K_{\text{init}}$ is set higher because, in the first step, $c$ does not have a warm start, and more iterations are needed to find reasonable initial values. $K_{\text{few}}$ is set to 2 as a trade-off between accuracy and computational cost (the comparison of costs is shown in Table 15). The selection of $t_c$ is based on observing when the change in $c$ stabilizes.
>
> Regarding the delayed residuals, the principle for setting $t_d$ and $t_r$ is to ensure that the network roughly satisfies the boundary conditions before the PDE residual term is introduced. We conducted an additional experiment in the Poisson benchmark to demonstrate performance under different $t_d$ and $t_r$ settings (Poisson + MLP, 5 seeds).
>
> | $t_d$/$t_r$ | 0/0 | 40/160 | 80/320 | 120/480 | 160/640 | 200/800 | 800/3200 |
> | --- | --- | --- | --- | --- | --- | --- | --- |
> | Stp | - | 5521 | 4713 | 5595 | 4574 | 3868 | 1984 |
> | $L_2$ | - | 9.99e-3 | 6.25e-3 | 1.00e-2 | 6.98e-3 | 5.00e-3 | 2.98e-3 |
>
> Overall, we found that for strong nonlinear BC, the later the residuals are introduced, the more stable the training becomes. Nevertheless, excessive $t_d$ may lead to overfitting of the BC, thereby causing gradient explosion after the introduction PDE residual. Therefore, we suggest set this value with greater caution.
>
> > **[Q1]** Why we choose Newton's method.
>
> As mentioned earlier, Newton's method has a quadratic convergence rate in convex optimization. This can significantly reduce the number of iterations. Although gradient descent only requires the calculation of first-order auto-diff in each run, it requires far more iterations than the Newton's method to achieve a same accuracy.
>
> Regarding the optimization methods you suggested, we conducted a set of additional experiments to compare its effectiveness (NS + MLP, 5 seeds). Among them, Broyden's method degenerates into the Secant method for 1D problems, which does not utilize the exact first-order derivatives available via auto-diff, making standard Newton's method a more direct and efficient choice.
>
> | | Stp | $L_2$ | $t_\text{alg}$ (s) |  Percentage |
> | --- | --- | --- | --- | --- |
> | Newton | 1269 | 4.73e-3 | 8.31 | 18.81% |
> | GD | 1506 | 4.99e-3| 4.69 | 10.23% |
> | Steffensen | 1375 | 4.98e-3 | 8.18 | 18.97% |
>
> Where the definitions of Stp and $L_2$ are the same as those shown in Table 2 in our main text, $t_\text{alg}$ is the time required for calculating $c$, and Percentage is the percentage of this period in the total training time. We will incorporate these methods into the discussion of the paper, and suggest that readers make more flexible choices based on the type of PDE. Thank you for your professional recommendation!
>
> > **[Q2]** Significant figures for SD.
>
> Thank you for the suggestion! We will change it to scientific notation during the revision.
>
> ---
>
> We promise to incorporate the above discussion into an appropriate section of our paper. Again, thank you for your recognition of our work!

---

> > ### Author Rebuttal · Reviewer_GUyz · 2026-04-06
> >
> > The main weaknesses and questions have been addressed.

---

> > > ### Author Response · Authors · 2026-04-06
> > >
> > > We once again thank you for your careful review based on the objective facts in our paper. It is truly our honor to have reviewers who are professional, meticulous and technically neutral!

---

### Decision · Program_Chairs · 2026-04-30

**Decision:**

Accept (regular)

**Comment:**

Most reviewers have identified substantial merit and novelty in the contributions of the work. While there were several comments and questions regarding the presentation, theoretical components, as well as the numerical evidence, the authors have addressed many of these concerns during the rebuttal to the satisfaction of most reviewers.

A number of concerns remain; however, these largely reflect individual perspectives and subjective judgments rather than concrete, objective scientific issues. Nonetheless, the work requires some revision, as discussed in the rebuttal, prior to being considered publication ready. In particular, I strongly encourage the authors to incorporate, as fully as possible, the remaining reviewer comments, along with the clarifications and changes proposed during the rebuttal, into the next version of the paper.